# Restoration of breathing after opioid overdose and spinal cord injury using temporal interference stimulation

Michael D. Sunshine [1,2,3,4], Antonino M. Cassarà [5], Esra Neufeld[5], Nir Grossman[6,7], Thomas H. Mareci [8], Kevin J. Otto[9,10,11,12,13], Edward S. Boyden [14,15] & David D. Fuller [2,3,4✉]

Respiratory insufficiency is a leading cause of death due to drug overdose or neuromuscular disease. We hypothesized that a stimulation paradigm using temporal interference (TI) could restore breathing in such conditions. Following opioid overdose in rats, two high frequency (5000 Hz and 5001 Hz), low amplitude waveforms delivered via intramuscular wires in the neck immediately activated the diaphragm and restored ventilation in phase with waveform offset (1 Hz or 60 breaths/min). Following cervical spinal cord injury (SCI), TI stimulation via dorsally placed epidural electrodes uni- or bilaterally activated the diaphragm depending on current and electrode position. In silico modeling indicated that an interferential signal in the ventral spinal cord predicted the evoked response (left versus right diaphragm) and current-ratio-based steering. We conclude that TI stimulation can activate spinal motor neurons after SCI and prevent fatal apnea during drug overdose by restoring ventilation with minimally invasive electrodes.

[1] Rehabilitation Science PhD Program, University of Florida, Gainesville, FL 32611, USA. [2] Department of Physical Therapy, University of Florida, Gainesville, FL 32611, USA. [3] Breathing Research and Therapeutics Center, University of Florida, Gainesville, FL 32611, USA. [4] McKnight Brain Institute, University of Florida, Gainesville, FL 32611, USA. [5] Foundation for Research on Information Technologies in Society (IT'IS), 8004 Zurich, Switzerland. [6] Division of Brain Sciences, Imperial College London, London SW7 2BU, United Kingdom. [7] United Kingdom Dementia Research Institute, Imperial College London, London SW7 2BU, United Kingdom. [8] Department of Biochemistry and Molecular Biology, University of Florida, Gainesville, FL 32611, USA. [9] J. Crayton Pruitt Family Department of Biomedical Engineering, University of Florida, Gainesville, FL 32611, USA. [10] Department of Neuroscience, University of Florida, Gainesville, FL 32611, USA. [11] Department of Neurology, University of Florida, Gainesville, FL 32611, USA. [12] Department of Materials Science and Engineering, University of Florida, Gainesville, FL 32611, USA. [13] Department of Electrical and Computer Engineering, University of Florida, Gainesville, FL 32611, USA. [14] Departments of Brain and Cognitive Sciences, Media Arts and Sciences, and Biological Engineering, McGovern and Koch Institutes, MIT, Cambridge, MA 02139, USA. [15] Howard Hughes Medical Institute, Cambridge, MA 02138, USA. ✉email: dfuller@phhp.ufl.edu

The absence of respiratory efforts (i.e., central apnea) after opioid overdose can be fatal and is a major component of the ongoing opioid abuse public health crisis[1]. Current treatments for rapidly restoring breathing after overdose include drug administration[2] and manual lung inflation. Non-invasive electrical stimulation technologies for rapid restoration of breathing are unavailable, but would be important for several reasons. First, a swiftly implementable method capable of immediately restoring breathing could sustain life while first responders cope with complex on-site medical, social or physical conditions. In turn, this could provide time for the first responder to implement additional restorative therapies. Second, when a drug such as naloxone is given to antagonize opioid receptors and revive an overdosed patient, withdrawal symptoms can be severe including seizures and heart rate irregularities[3]. In addition new synthetic opioids are more resistant to pharmacological reversal agents[4,5]. Direct electrical stimulation of breathing could avoid these problems. Third, simple and non-invasive electrical stimulation methods could provide health care professionals with an alternative or backup solution for restoring breathing if current approaches are ineffective, or could provide a user with no or limited medical training an option for restoring breathing efforts. For these reasons, we aimed to determine if the principles of temporal interference (TI)[6] could be used to immediately restore diaphragm muscle activity sufficient to restore ventilation and sustain life following opioid overdose. TI stimulation uses two high frequency waveforms (e.g., ≥2 kHz) that are slightly offset from each other (e.g., 1–10 Hz). This approach can drive activation of neurons located at considerable distance from the electrodes. The activation occurs at the offset or beat frequency of the two high frequency waveforms and with minimal stimulation of overlying structures[6]. Our first hypothesis was that that TI stimulation delivered via rapidly and easily placed wires in the neck region could generate rhythmic diaphragm activation during opioid-induced respiratory apnea, and thereby restore ventilation and arterial blood oxygenation.

Spinal cord injury (SCI) is another condition in which respiratory insufficiency can be deadly[7]. When hypoventilation is severe, mechanical ventilation[7] and phrenic nerve pacing[8] are medical options. Spinal epidural stimulation is another method capable of activating the paralyzed diaphragm[9–13]. This approach works if stimulating leads are placed on the ventral surface of the spinal cord, activating spinal motor neurons through an unknown neural substrate. Beyond breathing, epidural stimulation is also gaining considerable traction as a means of restoring somatic or autonomic function after SCI[14,15]. The state-of-the-art is to place an electrode over the spinal dura for delivery of a single electrical waveform[15–18] which may raise the excitability of spinal neurons[16,19] or possibly activate dorsal root sensory neurons[20], but does not appear to directly activate spinal neurons. Consequently, our goal was to determine if TI stimulation delivered via dorsally placed epidural electrodes could be used to activate motor neurons deep within the ventral spinal cord. Our second hypothesis was that TI electric fields could be steered to the ventral spinal cord to directly activate phrenic motor neurons, thereby restoring activity in the paralyzed diaphragm after cervical SCI. To this end, in vivo neurophysiological and pharmacologic methods in rats were complimented with in silico computational modeling based on electromagnetic simulations and response functions obtained from the neurophysiological data.

Collectively, the results of these experiments demonstrate that the interferential signal created by two high frequency waveforms can activate the phrenic neuromuscular system to (1) sustain breathing after opioid overdose, preventing fatal apnea or (2) activate spinal motor neurons following chronic cervical SCI. Thus, TI stimulation is a novel approach that restores breathing, and translation of this technology to clinical application may have value in a range of conditions associated with inadequate respiratory muscle activation. Beyond treatment of breathing, TI stimulation for targeted activation of neuronal populations at distance from the electrodes could revolutionize treatments for a range of neurologic disorders[21].

## Results

**Intramuscular TI stimulation activates the diaphragm.** We first determined if TI stimulation, delivered via rapidly placed intramuscular wires in the neck region (Supplementary Fig. 1) could robustly activate the diaphragm (Fig. 1a). When a 5 kHz carrier frequency was used with an offset/beat frequency of 1 Hz (Fig. 1b), robust diaphragm contractions were induced at 60 times per minute (Fig. 1c). To confirm that the observed diaphragm activation was resulting from TI, we also stimulated with high frequency kilohertz (HF, 5001 Hz, no offset) or low frequency (LF, 1 Hz) waveforms. These waveforms were unable to induce phasic diaphragm activation in any of the animals tested (Fig. 1d). Conversely, the TI stimulation paradigm activated the diaphragm in all animals tested (Fig. 1d), and the magnitude of the TI-evoked diaphragm burst was considerably greater than the spontaneously occurring EMG burst associated with inspiration (range: 270–931% increase). We next studied rats after opioid overdose-induced respiratory depression, which is the cause of mortality in the ongoing opioid epidemic[4,22]. Intravenous fentanyl (30 mcg/kg) produced respiratory apnea during which the diaphragm ceased rhythmic inspiratory contractions and arterial blood pressure decreased. Upon apnea, intramuscular TI stimulation was initiated and immediately produced robust diaphragm contraction with restoration of inspiratory airflow (Fig. 1e).

The TI rescue paradigm was initiated using 5000 + 5001 Hz waveforms shortly after the onset of opioid-induced apnea. Stimulation was delivered in 60 s epochs interspersed with brief 3 s off periods which were used to determine if endogenous diaphragm electromyography (EMG) activity had resumed (Fig. 2a, Supplementary Movie 1). The TI rescue effectively sustained breathing throughout the period of opioid-induced respiratory depression, thus animals were eventually able to resume independent (volitional) breathing after 28 ± 9 min of stimulation (range 17 to 40 min). Figure 2b and c illustrate the robust impact of TI stimulation on inspiratory airflow and arterial blood pressure during the period of opioid-induced respiratory depression, and this stands in sharp contrast to the no-intervention control group. Prior to opioid overdose, the respiratory rate ranged from 99–108 breaths per minute across the three experimental groups; all rats became apneic following intravenous fentanyl (i.e., 0 breaths per minute). The 1 Hz TI beat frequency was able to drive the respiratory rate during stimulation at 60 breaths per minute. When the TI beat frequency was increased to match the endogenous breathing rate (Matched freq.), diaphragm contractions occurred at an average 110 breaths per minute (Fig. 2d). This latter TI paradigm was sufficient to prevent arterial hypoxemia and also maintained arterial carbon dioxide values within normocapnic ranges. In contrast, TI stimulation which evoked a lower rate of diaphragm contraction (60 breaths per minute) was sufficient to maintain arterial oxygenation but did not fully correct hypoventilation induced arterial hypercapnia (Supplementary Fig. 2). The survival curves shown in Fig. 2e serve to highlight the effectiveness of intramuscular TI stimulation for sustaining breathing, as well as the potential value of this approach in a first responder situation following opioid overdose.

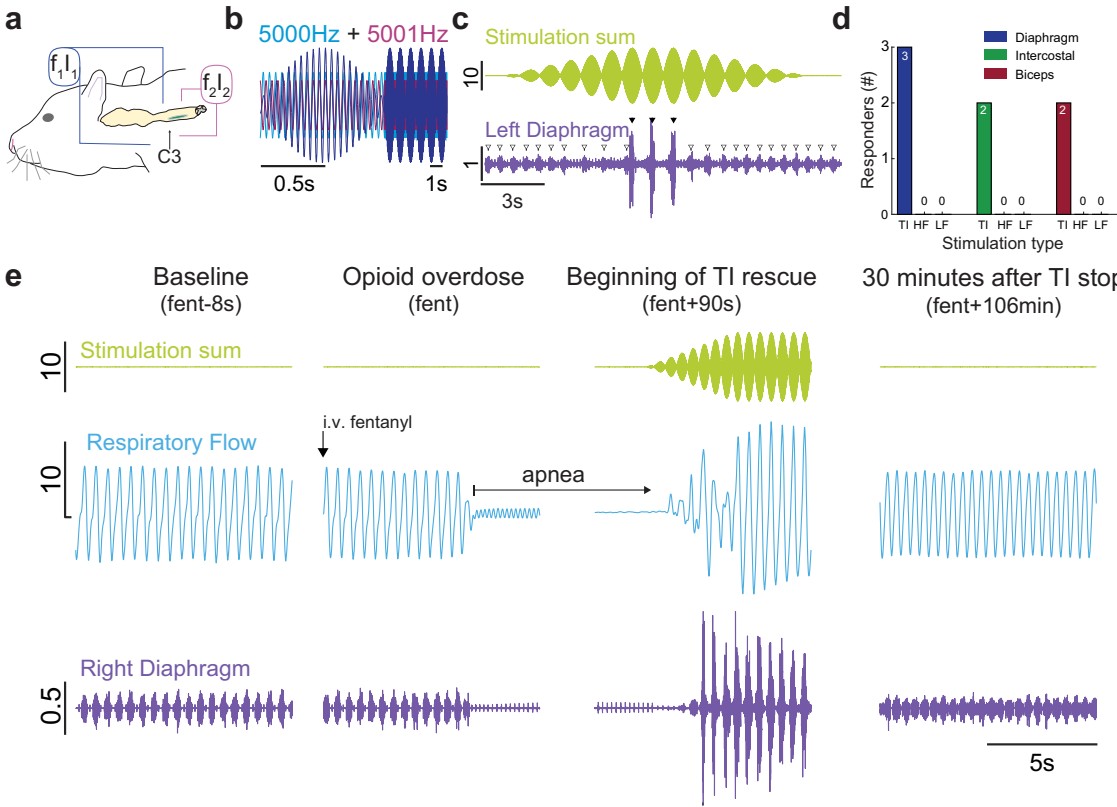

**Fig. 1 Representative traces show that intramuscular TI stimulation restores diaphragm EMG activity and inspiratory airflow after opioid overdose. a** Schematic of wire placement targeting C3 to activate the phrenic motor pool (green column). Stimulation was delivered via a pair of wires on each side of the neck. One wire per side was inserted into the neck musculature with intent to target the ventral and dorsal aspect of the spinal column near C3. The left electrode pair (blue) was stimulated with current-1 ($I_1$) and frequency-1 ($f_1$), and the right pair (pink) was stimulated with current-2 ($I_2$) and frequency-2 ($f_2$). **b** Stimulation paradigm. $f_1 = 5000$ Hz (blue), $f_2 = 5001$ Hz (pink), produces a TI field with a modulation envelope or beat frequency of 1 Hz (purple), which produced a respiratory rate of 60 breaths per minute. **c** TI stimulation causes robust diaphragm EMG activation in phase with the TI beat frequency (shown in green). TI-induced stimulation (black triangles) was able to activate the diaphragm well above levels seen during the spontaneously breathing baseline period (white triangles). **d** Temporal interference (TI) stimulation results in robust phasic activation of the diaphragm in $n = 3/3$ animals, while control high frequency (HF), and low frequency (LF) waveforms do not activate the diaphragm (Fisher's exact test, diaphragm: $p = 0.0119$, intercostal: $p = 0.083$, bicep: $p = 0.083$). **e** Representative data from a single animal showing right hemi-diaphragm activity and respiratory airflow during spontaneous breathing (baseline, left panel), cessation of diaphragm inspiratory EMG activity following opioid overdose (apnea, middle-left panel), restoration of EMG activity and respiratory airflow via TI stimulation (middle-right panel), and resumption of spontaneous EMG activity and respiratory airflow lasting 30 min beyond the TI stimulation period (right panel). Respiratory airflow at baseline occurs in phase with spontaneous diaphragm EMG bursts. Intravenous fentanyl stopped respiratory airflow (remaining oscillations reflect cardiac pressures). TI rescue activated the left diaphragm in phase with the TI beat frequency (illustrated by the green trace). In this example, TI stimulation activated the left diaphragm well above baseline values but did not activate the right diaphragm. Left diaphragm activation was sufficient to sustain life until opioid-induced respiratory depression was no longer present. Scale bar units: stimulation (mA), diaphragm (mV), and respiratory flow (ml/s).

**Epidural TI stimulation effectively activates the diaphragm.** We next asked if we could activate ventral (phrenic) motor neurons using dorsally placed epidural electrodes, as these do not require piercing the spinal parenchyma, and if stimulation waveforms could be manipulated to steer the current towards the left or right spinal cord. This experimental preparation also enabled us to determine if spinally-directed TI stimulation requires presynaptic inputs to motor pools to be effective. This was accomplished by examining TI stimulation-evoked diaphragm activity following local delivery of molecules which blocked excitatory and/or inhibitory synaptic inputs to spinal motor neurons.

Initial epidural stimulation experiments were conducted in spinal-intact rats using bipolar, as well as monopolar electrode configurations (Fig. 3a). Stimulating neural tissue with bipolar electrode configurations allows for electrical current to move between two electrode pairs, thereby producing a relatively localized current flow. Monopolar electrode configurations

provide for flow of electrical current through a single electrode that is referenced to a distant return. The ratio of TI stimulation-evoked diaphragm EMG activity to the spontaneous inspiratory EMG burst, evoked ratio (e.g., Fig. 3b) was used to verify that stimulation produced phasic EMG bursts in time with the TI envelope (i.e., periodic stimulation at the difference frequency). This analysis showed that bipolar electrode configurations did not robustly activate the diaphragm whereas monopolar configurations evoked large phasic bursts in phase with the interference envelope (Fig. 3c, d). Further, stimulating via the electrodes over the C3 segment had the least impact on the biceps and external intercostal (Fig. 3d), and thus subsequent experiments focused on C3 electrode placement with a monopolar configuration. To assess if the location of the common return was important, the return location was systematically varied using a low impedance 127 µm tungsten wire implanted in either the back muscles (dorsal and distal), ventral neck muscles (ventral and distal), or paraspinal muscles (lateral and proximal). Varying the return

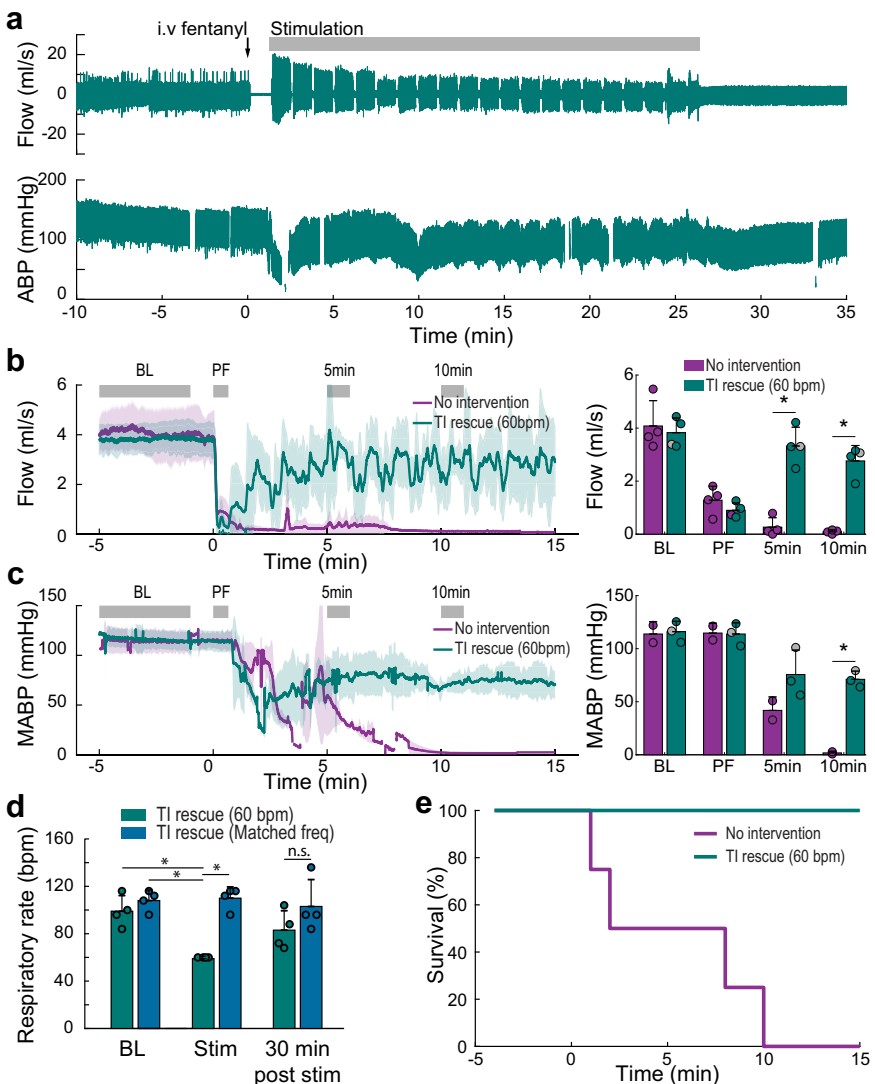

**Fig. 2 Intramuscular TI stimulation (TI rescue) prevents fatal apnea by restoring diaphragm activation after opioid overdose.** Spontaneously breathing rats under urethane anesthesia received an intravenous injection of fentanyl (30 mcg/kg) sufficient to cause lasting respiratory suppression. Following fentanyl dosing, animals received intramuscular TI stimulation or no intervention. Panel **a** shows an example of sustained restoration of respiratory airflow when the stimulation was initiated shortly after opioid overdose-induced apnea, and was delivered with a pattern of 60 s on and 3 s off until spontaneous breathing resumed. Note: The gaps in the blood pressure traces indicate when arterial blood samples were taken. **b** Average respiratory airflow (two-way ANOVA: group F(1) = 40.84, $p < 0.0001$; time F(3) = 41.86, $p < 0.0001$; interaction F(3) = 21.4, $p < 0.0001$) and (**c**) mean arterial blood pressure (MABP) ($n = 4$ animals/group, two-way ANOVA: group F(1) = 19.81, $p = 0.0008$; time F(3) = 45.95, $p < 0.0001$; interaction F(3) = 7.82 $p = 0.0037$). The gray bars indicate time periods selected for quantitative comparison (Baseline–BL, immediately Post Fentanyl–PF, five minutes–5 min, and ten minutes–10 min after fentanyl). The gray dots indicate data points from the animals used for the example traces in panel a, (mean + 1 standard deviation; asterisk (*) indicates $p < 0.05$). **d** TI stimulation restored ventilation and prevented fatal apnea in all animals (TI rescue). TI stimulation with a 1 Hz beat frequency produced a respiratory rate of 60 breaths per minute (bpm) during stimulation. When the beat frequency was targeted to the endogenous respiratory rate, there was no difference in the evoked vs. spontaneous respiratory rate ($n = 4$ animals, two-way ANOVA: group F(1) = 22.64, $p = 0.0002$; time F(3) = 3.85, $p = 0.0407$; interaction F(3) = 5.03, $p = 0.0183$). **e** Survival curve for no-intervention ($n = 4$ animals) and TI rescue (60 bpm) conditions ($n = 4$ animals).

location in this manner had no discernable impact on the ability to activate the diaphragm (Fig. 3e). To verify this effect, we modeled the impact of the location of the return electrode in accordance with the experimental data shown in Fig. 3e. For that purpose, the return electrode was rotated in six positions around the ventral side of the spinal cord, while maintaining a constant inter-electrode distance. This modeling produced a variability of ±3%, and thus the simulation results confirm that the location of the return electrode had minimal impact on diaphragm activation. To confirm that TI was required for in vivo diaphragm activation with this electrode configuration, we tested the effect of

high frequency kilohertz waveforms with no offset (HF, 5001 Hz) or low frequency (LF, 1 Hz) waveforms on diaphragm activation. Neither of these waveforms were able to cause phasic diaphragm activation similar to the response to TI (Fig. 3f).

We next tested if epidural TI stimulation could activate the paralyzed diaphragm after cervical spinal cord injury (schematic shown in Fig. 3g). We utilized a high cervical (C2) hemilesion model (C2Hx) which results in transient paralysis and chronic paresis of the ipsilateral hemi-diaphragm (Fig. 3h, left panel). Epidural TI stimulation using a C3 monopolar configuration robustly activated the previously paralyzed hemi-diaphragm as

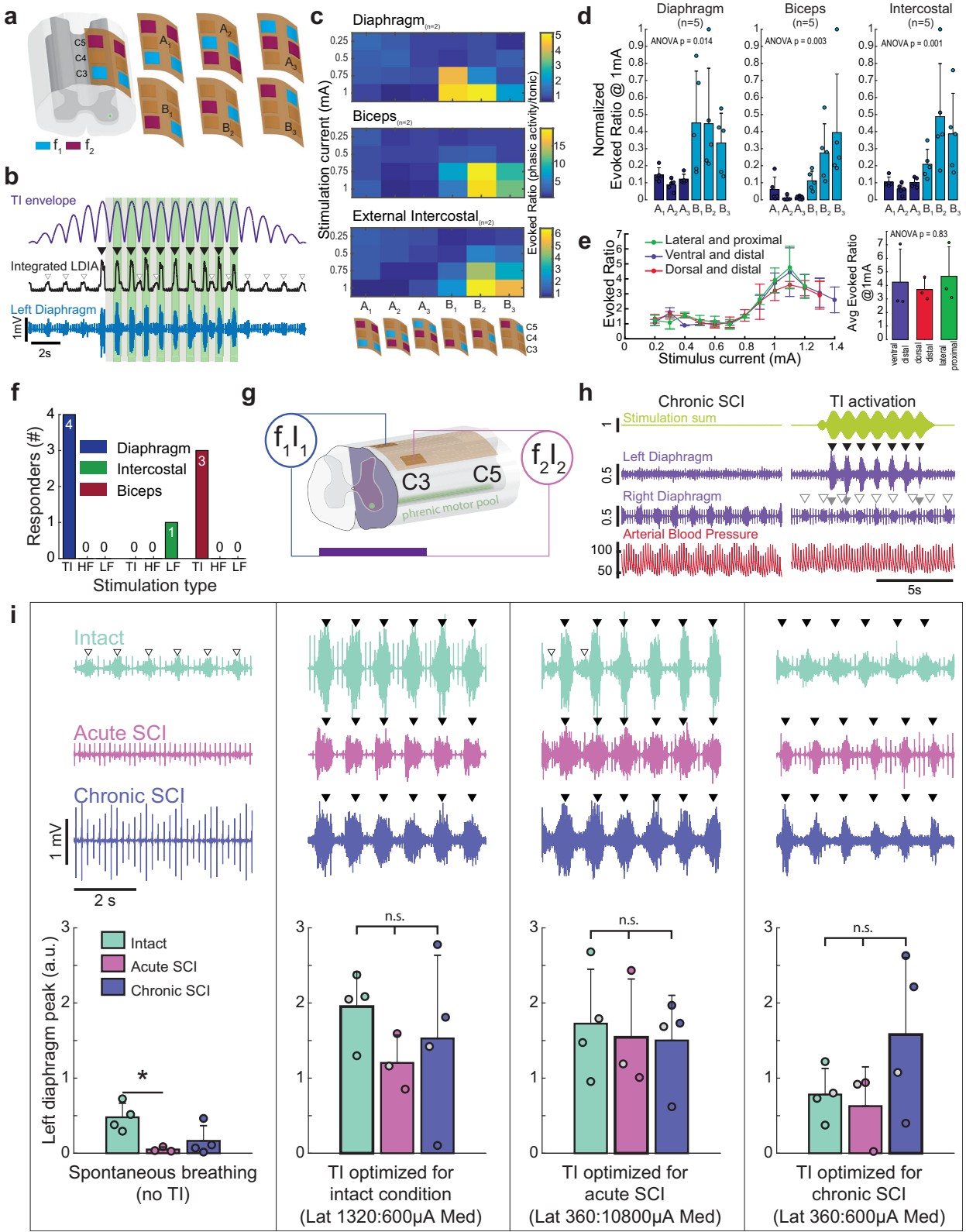

evidenced by large EMG bursts occurring in phase with the TI beat frequency (Fig. 3h, right panel). We systematically evaluated this response after acute or chronic (10 month) spinal cord injury in separate experimental cohorts. The C2Hx lesion immediately eliminated spontaneous EMG activity in the ipsilateral diaphragm in the acutely injured rats and caused a substantial reduction in spontaneous EMG output in those with chronic injury (Fig. 3i).

For each condition (e.g., spinal intact, acute and chronic injury), we determined the stimulus current ratio (i.e., ratio of the two TI currents) that evoked the largest peak diaphragm EMG output. Figure 3i shows a summary of the evoked diaphragm EMG response at these empirically determined optimal currents. Collectively, these data show that epidural TI stimulation effectively activates the diaphragm after spinal cord injury. The

**Fig. 3 Effect of electrode configuration on diaphragm activation and diaphragm response to epidural TI stimulation. a** Initial testing to determine left diaphragm (LDIA), bicep, and external intercostal activation utilized an epidural electrode grid spanning C3–C5 and with three bipolar (A1, A2, and A3) and three monopolar configurations (B1, B2, and B3); note: electrode size is exaggerated for schematic, actual wire width is 25 μm. **b** The ratio of TI stimulation-evoked EMG output to spontaneous EMG activity was used to quantify the response. The EMG activity during the peak of the TI envelope (green shaded boxes) was divided by the activity in the trough of the envelope (gray shaded boxes) for the period of stable stimulation (i.e., excluding the ramp and damp phases). **c** Heat maps which display the activation of the diaphragm, biceps, and external intercostal muscles as a function of stimulus amplitude and electrode configuration (waveform current ratio 1:1 in these examples). The particular electrode configuration is shown on the bottom of the panel. **d** The average evoked diaphragm EMG activity (mean + 1 standard deviation) during TI stimulation using 1 mA in both waveforms (asterisk (*) indicates $p < 0.05$ tukey's post-hoc). Electrode configuration B1 (C3 monopolar stimulation) activated the diaphragm with minimal off-target biceps activation ($n = 5$ animals, Kruskal–Wallis one-way ANOVA on ranks: diaphragm H(5) = 20.73, $p < 0.001$; biceps H(5) = 22.83, $p < 0.001$; intercostal H(5) = 23.09, $p < 0.001$). **e** Effect of varying the location of the current return on diaphragm activation during C3 monopolar TI stimulation. Three different current return electrode locations were used (RM two-way ANOVA current F(26) = 8.79, $p < 0.001$; return location F(2) = 2.51, $p = 0.092$; $n = 3$ animals). **f** Temporal interference (TI) stimulation phasically activated the diaphragm in $n = 4/4$ animals, while high frequency (HF), and low frequency (LF) waveforms did not phasically activate the diaphragm (Fisher exact test; diaphragm, $p = 0.002$; intercostal, $p = 0.333$; bicep, $p = 0.018$). **g** Schematic of the mid-cervical spinal cord injury (purple indicates spinal hemilesion) and electrode locations. **h** Example data from a rat with chronic (10 months) cervical spinal cord injury. At baseline (without stimulation, left panels), the left hemi-diaphragm is inactive while the right hemi-diaphragm shows rhythmic bursting. Electrocardiogram (ECG) activity is present in both traces. TI stimulation (right panel) immediately activates the left hemi-diaphragm (black arrowheads) and produces small bursts in the right diaphragm (gray arrowheads). Spontaneous activity in the right hemi-diaphragm (white arrowheads) is uninterrupted. Units: stimulation (mA), diaphragm (mV), arterial blood pressure (mmHg). Panel **i** provides additional examples of diaphragm EMG output and mean responses. The left panel shows diaphragm EMG during spontaneous breathing in spinal intact, acute, and chronic spinally injured rats (mean diaphragm bursting is shown in the plots at the bottom of the panel). Proceeding left to right across the figure, evoked EMG responses are shown using stimulus current ratios optimized for each condition (i.e., spinal intact, acute and chronic injury). Spontaneous bursts (white arrowheads) are present in the spinal intact animal but are absent after acute and chronic SCI. TI stimulation effectively activates the diaphragm (black arrowheads) in all three conditions. Plots: one way ANOVA spontaneous breathing F(2) = 6.186, $p = 0.024$; optimized for intact F(2) = 0.863, $p = 0.458$; optimized for acute SCI F(2) = 0.115, $p = 0.893$; optimized for chronic SCI F(2) = 1.914, $p = 0.209$; asterisk (*) indicates $p < 0.05$ tukey's post-hoc; gray filled dots indicate animals used in example traces (intact, $n = 4$ animals; acute SCI, $n = 3$ animals; chronic SCI, $n = 4$ animals).

afferent and efferent innervation of the phrenic motor nucleus, and the local tissue architecture, will be different when comparing acute to chronic spinal cord injury. TI stimulation was effective at both time points, indicating that time-dependent changes in the spinal cord after injury do not impair the effectiveness of this method.

**Current steering in the spinal cord.** One of the theoretical benefits of TI stimulation is the ability to shift the focal point where the electric fields overlap (Supplementary Fig. 3) to activate neurons deep within the target tissue and distant from the electrodes[6]. To determine if the focal point in the cervical spinal cord could be steered during epidural TI stimulation, we varied the ratio of the two current sources while maintaining a constant total current. This experiment was done using electrodes placed laterally or spanning the dorsal surface of the spinal cord. Figure 4a shows that TI stimulation delivered using lateral electrode placement (1.5 mm inter-electrode distance) produced rhythmic activation of the ipsilateral (left) but not contralateral (right) diaphragm at the TI beat frequency. When the total current across the two electrodes was held constant at 1800 μA, ratios of 1:1, 1:2, and 1:3 produced the largest phasic diaphragm activation. When total current was 2400 μA, ratios of 3:1, 2:1, and 1:1 produced the largest phasic diaphragm activation. Figure 4b shows the results of the TI current steering experiment using the bilateral electrode configuration (3 mm inter-electrode distance). In contrast to the results with lateral placement, the bilateral configuration tended to produce bilateral diaphragm activation. Current steering was still present as ratios of 1:1 and 1:2 were most likely to produce bilateral diaphragm activation with total stimulus currents of 1800 and 2400 μA.

Both the lateral and bilateral electrode configurations evoked sustained and non-rhythmic diaphragm activity when the current ratios were 0:1 or 1:0 (i.e., a single high frequency waveform). The amplitude of this tonic diaphragm activation at 0:1 or 1:0 current ratios was attenuated as the stimulation progressed, and this can be seen in the example traces shown in Fig. 4a, b.

**Blocking synaptic inputs alters the TI stimulation response.** Inspiratory-related depolarization of phrenic motor neurons arises primarily from excitatory bulbospinal pathways in the ventrolateral white matter of the mid-cervical spinal cord[23]. There also exists a complex propriospinal network that can excite or inhibit phrenic motor neurons[24] (Fig. 5a). If the diaphragm activation resulting from TI stimulation is primarily due to depolarization of excitatory presynaptic inputs, focal delivery of glutamate receptor antagonists to the phrenic motor pool should substantially reduce or eliminate the evoked response. Conversely, if the TI stimulation paradigm is also activating inhibitory synaptic inputs, then antagonizing inhibitory receptors in the phrenic motor pool should increase the diaphragm response. To test these possibilities, the epidural electrode configuration was modified to enable intraspinal microinjections without moving the electrode (Fig. 5b). After establishing a baseline TI response, we then intraspinally injected a mixture of glutamatergic receptor antagonists to block NMDA (AP5), and non-NMDA (CNQX) receptors, or a strychnine/bicuculline mixture to block glycine and GABA$_A$ receptors, respectively. Each group was then subsequently injected with the other drug mixture (Fig. 5c). Delivering CNQX/AP5 to the spinal cord almost completely eliminated endogenous (spontaneous) inspiratory diaphragm EMG bursts (Fig. 5d). In contrast, strychnine/bicuculline had little to no effect on the amplitude of endogenous bursts (Fig. 5e). Following application of both drug cocktails there was near complete elimination of endogenous diaphragm EMG bursts (Fig. 5f). The response to TI stimulation after the spinal drug injection was evaluated at the medial stimulus current which evoked an EMG burst that was 25% larger than endogenous activation. Blocking local spinal excitatory neurotransmission substantially reduced diaphragm activation during TI stimulation (Fig. 5g$_i$), whereas blocking inhibitory neurotransmission had minimal impact (Fig. 5g$_{ii}$). The mean data shown in Fig. 5h demonstrate that following spinal delivery of CNQX/AP5 the evoked diaphragm response was decreased by 80, 58, and 54% at lateral currents of 100, 200, and 300 μA, respectively. The subsequent delivery of

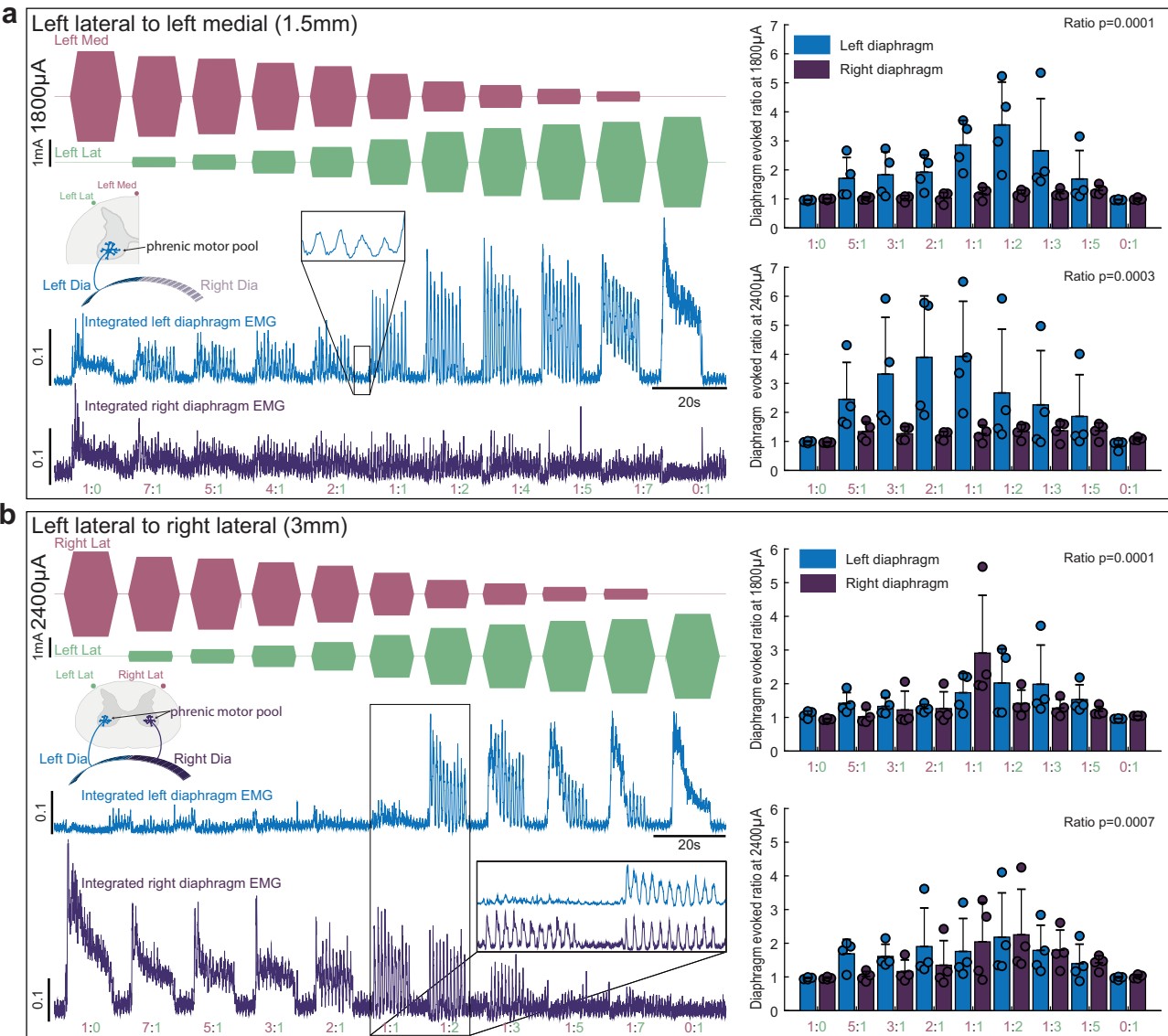

**Fig. 4 Current steering during epidural TI stimulation.** Using diaphragm EMG as the outcome measure, the steerability of the TI focal point was examined by varying the ratio of the two stimulus currents using two inter-electrode distance. Stimulating electrodes were placed on the left side of the spinal cord (**a**, 1.5 mm apart) or spanning the dorsal surface (**b**; 3 mm apart). The left side of panel **a** shows an example of rectified and integrated diaphragm activity when TI was delivered with lateral electrode placement and a current sum of 1800 µA. In this example it can be appreciated that varying the TI current ratio dramatically alters the magnitude of phasic diaphragm activation, and tonic diaphragm activation occurs at current ratios of 1:0 and 0:1. The mean data (right panels) show a statistical effect of current ratio at currents sums of 1800 µA (Friedman's two-way ANOVA controlling for diaphragm side effect of current ratio Chi-squared(8) = 33, $p < 0.0001$) and 2400 µA (Friedman's two-way ANOVA controlling for diaphragm side effect of current ratio Chi-squared(8) = 23.96, $p = 0.0023$). The left side of panel **b** shows an example of diaphragm activity evoked by stimulating the spinal cord using an inter-electrode distance of 3 mm and a current sum of 2400 µA. In this example, TI stimulation induced a robust bilateral and phasic diaphragm EMG activation at a current ratio of 1:2. The mean data show an effect of current ratio at sum current of 1800 µA (Friedman's two-way ANOVA controlling for diaphragm side effect of current ratio Chi-squared(8) = 29.33, $p = 0.00003$) and at 2400 µA (Chi-squared(8) = 26.99, $p = 0.00007$). Note we lowered inspired $CO_2$ to reduce endogenous diaphragm activity during these trials, small bursts are still present (inset panel **a**), but at much lower amplitude than the TI-evoked bursts ($n = 4$ animals).

strychnine/bicuculline had no discernable additional impact on the TI response. When inhibitory neurotransmission was blocked first (i.e., before the excitatory blockade), there was also little impact on the TI-evoked diaphragm activation (Fig. 5i). Subsequent delivery of CNQX/AP5, however, caused a considerable attenuation of diaphragm activation during TI stimulation (Fig. 5i). Since local spinal blockade of excitatory synaptic inputs reduced, but did not eliminate, the response to TI stimulation; activation of presynaptic inputs to phrenic motor neurons is

implicated as a component of the mechanism driving TI-induced diaphragm muscle activation.

To demonstrate that diaphragm EMG activation was not the result of extracellular electric field propagation directly depolarizing the diaphragm myofibers, TI stimulation was repeated after intravenous delivery of a drug to block acetylcholine binding at the neuromuscular junction (pancuronium bromide, 2.5 mg/kg, i.v., Hospira). The diaphragm could not be activated by TI stimulation after pancuronium bromide, indicating that the

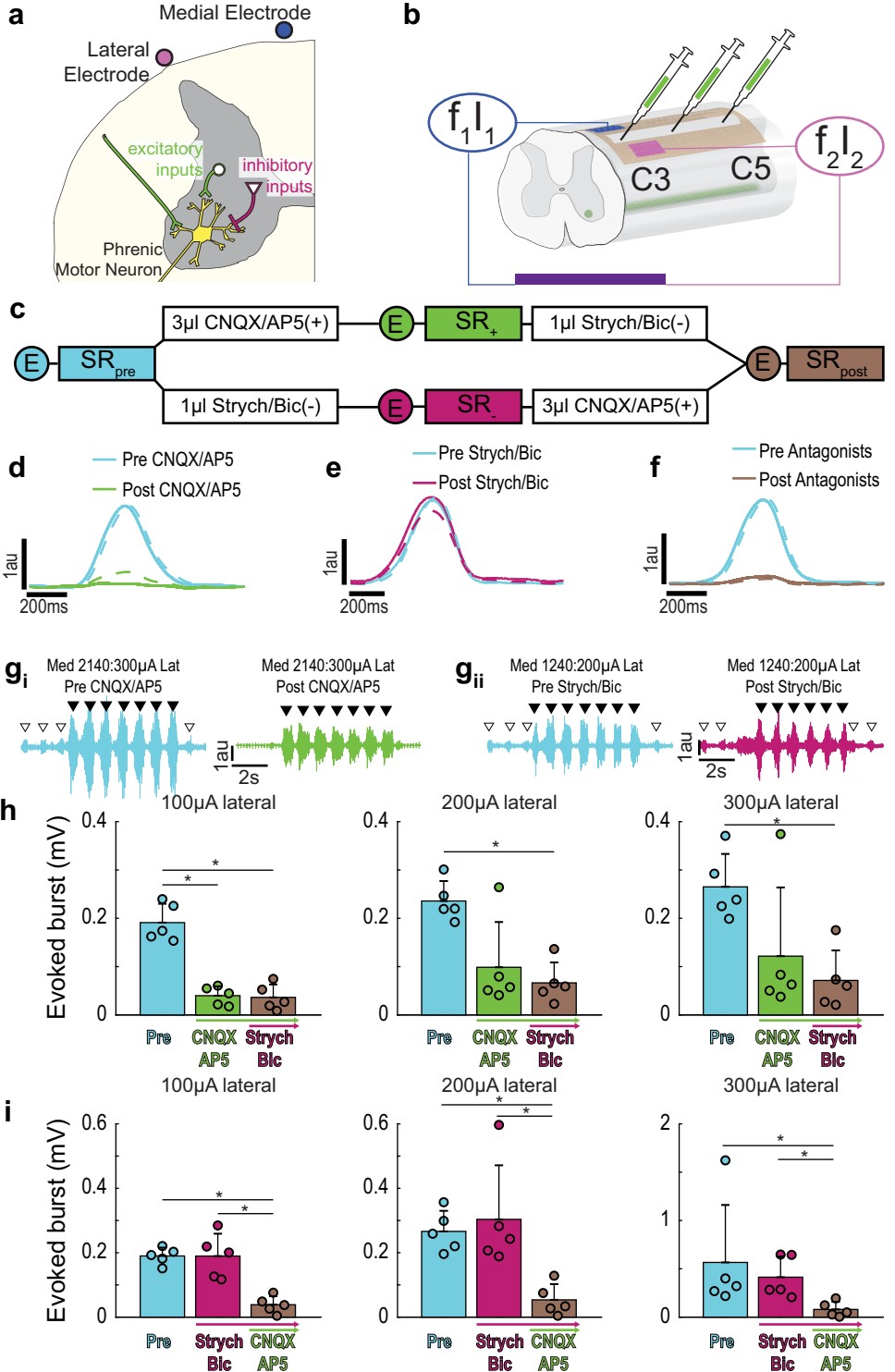

response requires activation of cholinergic synaptic inputs to the diaphragm (Supplementary Fig. 4).

**Stimulation using alternative waveforms.** Stimulation at kilohertz frequencies (e.g., 5001 Hz) with a single waveform did not induce diaphragm activity when applied intramuscularly (Fig. 1d, Supplementary Movie 2) or epidurally (Fig. 3f). Similarly, intramuscular or epidural stimulation with a sine wave at the beat frequency, with no carrier frequency (i.e., 1 Hz) did not activate the diaphragm. TI stimulation in these same animals elicited a robust diaphragm burst

(Supplementary Fig. 5a). Square wave stimulation at 50 pulses per second (pps) was able to elicit compound muscle action potentials in the diaphragm with concurrent biceps activation. These muscle action potentials did not summate into a sustained diaphragm burst as occurs during inspiration (Supplementary Fig. 5b). Square wave stimulation at 300 pps evoked a square wave diaphragm burst (Supplementary Fig. 5c) concurrent with large amplitude activation of the biceps muscle on the corresponding side. In addition, both the 50 pps and 300 pps stimulation caused widespread muscle activation throughout the rats.

**Fig. 5 Pharmacologically probing the contribution of synaptic input to phrenic motor neurons in TI stimulation-induced diaphragm activation.** To determine if TI stimulation activates the diaphragm due to direct depolarization of phrenic motor neurons or through activation of synaptic inputs to these cells, we utilized focal injections of glutamate, glycine, and GABA$_A$ receptor antagonists. **a** Diagram of the phrenic motor neuron pool illustrating excitatory (green) and inhibitory (pink) presynaptic inputs. **b** Schematic of the epidural stimulation grid and unilateral (left) intraspinal drug injection. **c** Summary of the experimental paradigm. In one cohort ($n = 5$ animals), excitatory presynaptic inputs were antagonized first, followed by inhibitory antagonists. In another cohort ($n = 5$ animals) the order was reversed. Circles with E indicate when endogenous activity was measured, SR indicate when stimulus response curves were performed. Panels **d–f** provide cycle triggered averages of endogenous diaphragm EMG activity (using the activity in the right (unblocked) hemi-diaphragm as the trigger). **d** Before (blue) and after (green) intraspinal injection of CNQX/AP5 ($n = 5$ animals). **e** Before (blue) and after (pink) intraspinal injection of strychnine/bicuculline ($n = 5$ animals). **f** Before (blue) and after (brown) both sets of drugs were injected ($n = 10$ animals). The solid line is average of breaths over the first 30 s post-injection and the dashed line is the average of the breaths over the last 30 s. **g$_i$** Example diaphragm EMG illustrating the magnitude of TI stimulation-evoked responses before (blue, pre CNQX/AP5) and after glutamate receptor antagonism (green, post CNQX/AP5). Endogenous bursts (white arrowheads) were eliminated after CNQX/AP5 indicating that excitatory drive to the phrenic motor pool was effectively blocked. Black arrowheads indicate the peak TI envelope, marking evoked bursts. **g$_{ii}$** Example diaphragm EMG illustrating the magnitude of evoked responses before (blue, pre-strychnine/bicuculline) and after glycinergic and GABAergic receptor antagonism (pink, post strychnine/ bicuculline). Stimulus current on the medial:lateral wires are shown above the traces. **h** Impact of CNQX/AP5 followed by strychnine/bicuculline on the TI-evoked diaphragm EMG burst. ($n = 5$ animals). **i** Impact of strychnine/bicuculline followed by CNQX/AP5 on the TI-evoked diaphragm EMG burst ($n = 5$ animals). In panels **h** and **i**, data are shown for 100, 200, and 300 μA lateral currents with medial current standardized at the value which evoked diaphragm EMG activity 25% above endogenous (pre-drug) baseline bursting. All data presented as mean +1 SD (*$p < 0.05$ post hoc, panel **h** at 100 μA: one-way ANOVA F(2) = 44.77, $p < 0.001$; all other panels: Kruskal–Wallis one-way ANOVA on ranks; panel **h** at 200 μA H(2) = 6.860, $p = 0.032$; panel **h** at 300 μA H(2) = 6.720, $p = 0.035$; panel **i** at 100 μA H(2) = 9.380, $p = 0.009$; panel **i** at 200 μA H(2) = 9.420, $p = 0.009$; panel **i** at 300 μA H(2) = 9.380, $p = 0.009$).

**Computational modeling successfully predicts current steering and respiratory response.** Simulation models of the electric tissue exposure have been established (Fig. 6a; see Experiment 5 in the "Methods" section), reproducing the asymmetric, 1.5 mm-distance and the symmetric, 3 mm-distance epidural stimulation experimental configurations from Fig. 4c, d. Based on the computed electric fields (Supplementary Fig. 3b), the exposure of potential candidate regions that could affect the respiratory response ('Motor Pool', 'Intermediate Gray Matter', 'Bulbospinal', and 'Afferent' pathways) was quantified in terms of the TI metric from Grossman et al.[6]: Simulated TI field distributions for a range of current steering ratios, as well as the associated average exposure of the four regions of interests can be seen in Fig. 6b, c. To translate exposure to evoked respiratory response, experimental data obtained by varying the current while maintaining a 1:1 current ratio (as shown in Fig. 3e) has been fitted (Fig. 6d; see Experiment 5 in the "Methods" section). By combining these response functions with the TI metrics extracted from the electromagnetic simulations it becomes possible to predict the evoked respiratory response for any set of current steering parameters. The differences between the predicted and the experimentally measured evoked ratios are in the range of 14–30% (relative to the average evoked ratio for a given total current; Fig. 6e). Figure 6f shows a direct comparison of the computational predictions with the experimental results from Fig. 4a, b. Considering that the experimental variability related to the biological response, intra-subject anatomical variability, and/or electrode placement reproducibility already accounts for 20%, which does not yet include the modeling uncertainty, it is concluded that computational modeling successfully predicts current steering and respiratory response. The 20% are estimated based on (1) the 21% standard deviation of the evoked responses of the left and right diaphragm for corresponding—i.e., mirrored—steering ratios in what should be a symmetric setup, see Fig. 4d, and (2) the average variability in the experimental data shown in Fig. 3e. The simplifications and limitations of the computational model are discussed in more detail in the "Methods" section.

**Computational modeling explains the counter-intuitive current-ratio-dependence of the respiratory response, confirms the relevance of the modulation envelope magnitude metric.** While it can at first be counter-intuitive that shifting the current ratio towards increased prominence of the right electrode (while maintaining the total current) results in a converse shift of the evoked activity to the left diaphragm and vice versa, this is in fact predicted by the computational model. It can be understood in light of the observation from Grossman et al.[6] that the amplitude of the modulation envelope predicts TI stimulation. For mostly aligned field orientations, the modulation envelope amplitude is simply twice the amplitude of the smaller of the two electric fields. For field strengths that decay with increasing distance from the electrodes, the maximal modulation envelope amplitude is therefore found between the electrodes, at locations where the two field strengths have a comparable magnitude. As the current ratio shifts towards the right electrode, the field generated by the left electrode decreases (proportionally to the left electrode current) and the location where the two fields are comparable and temporal interference is maximal, shifts towards the left side (Fig. 4). The different candidate regions are affected to a different degree by that shift (Fig. 6c) and it is suggested by the deviations between computational predictions and experimental data (Fig. 6f) that the TI exposure of the phrenic motor pool region is principally responsible for the evoked respiratory response, as it provides the most accurate predictions.

## Discussion

Breathing is fundamental to life and can be severely compromised in many neuromuscular diseases and injuries, as well as drug overdose. Impaired respiratory muscle activation is a leading contributor to morbidity and mortality across a range of conditions, and central apnea (i.e., absent neural drive to breathe) can be fatal if not treated immediately. Here we show the TI stimulation is a novel technology for rapidly restoring breathing after opioid overdose. Further, we demonstrated that TI stimulation provides a new experimental avenue in the rapidly growing area of epidural stimulation following spinal cord injury. Thus, TI is a promising new modality for respiratory stimulation.

Interferential current stimulation paradigms have been used in physical therapy since at least the 1950s[25,26]. Initially described by Nemec[27] as a transcutaneous stimulation method, TI stimulation has been suggested to have less superficial sensory impact compared to other stimulation modalities, and to be effective for reducing pain when applied with surface (skin) electrodes[25]. However, clinical studies have not provided support for this

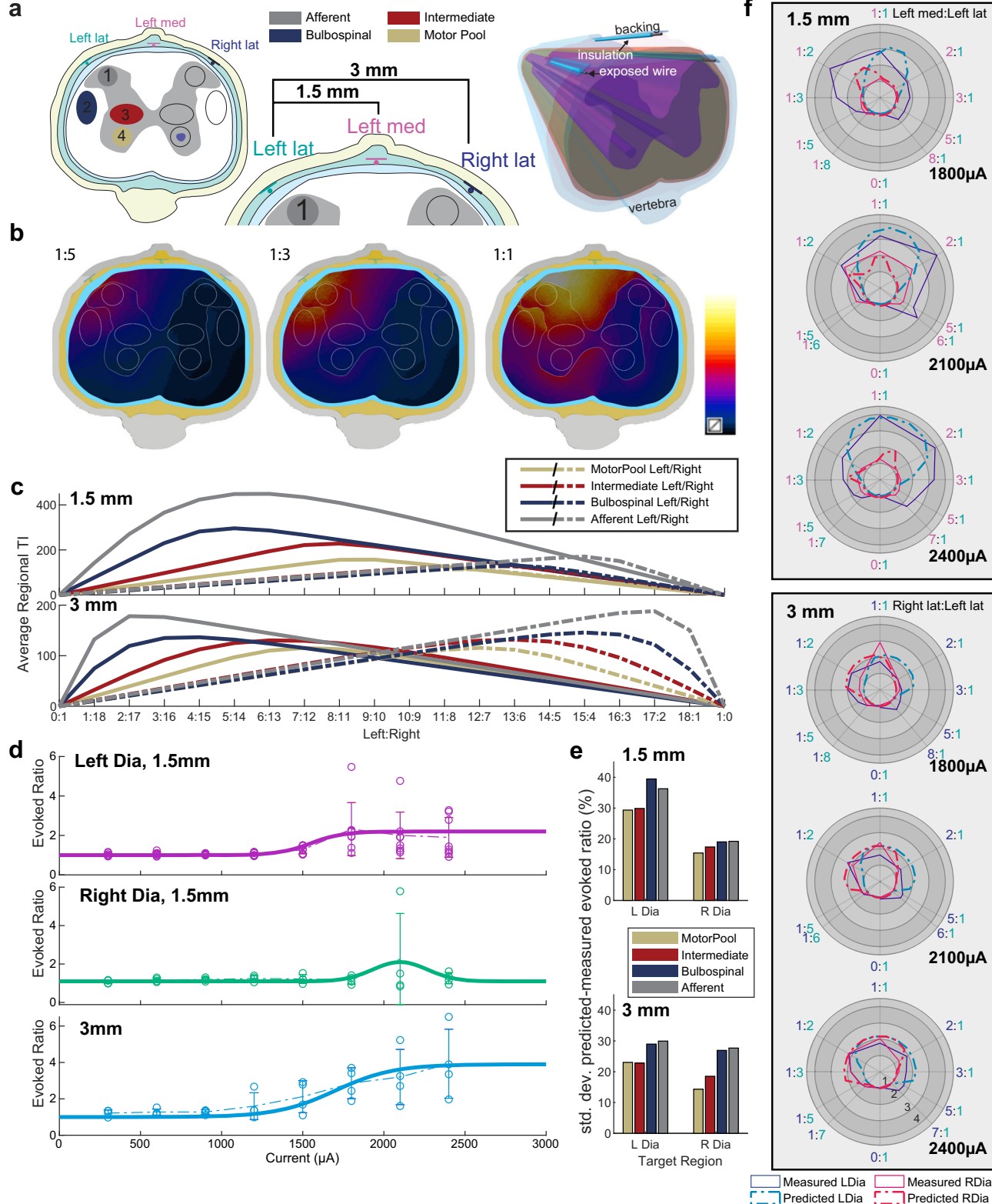

notion[28,29], and whether TI stimulus patterns are effective in physical therapy settings remains an open question.

Grossman and colleagues reasoned that TI stimulation could enable non-invasive activation of deep brain structures[6]. In a mouse model, TI stimulation delivered via electrodes at the brain surface convincingly demonstrated that TI stimulation could activate hippocampal neurons, without activation of overlying (cortical) structures. The TI currents were steerable within deep

brain structures and this was accomplished without physical movement of the stimulus electrodes. The low-pass filtering ability of neural membranes was suggested to explain the relative lack of impact of kilohertz electrical fields delivered without interference. That TI stimulation was necessary and sufficient for activation of neurons was validated by showing that (1) neurons were activated at the TI beat frequency (i.e., discharge rates follow the frequency envelope of the interfering electric fields, but not

**Fig. 6 Computational prediction of TI stimulation-induced evoked response and current steering. a** Simulation setup reproducing the experimental setup. Three electrodes are placed dorsally in the epidural space of the spine (left lateral, right lateral, and 'Left' medial), while a shared return electrode is placed ventrally. TI modulation amplitude maps are generated using two different configurations of channel pairs: the '1.5 mm' configuration using the left lateral and the medial electrodes, as well as the '3 mm' configuration using the left and the right lateral electrodes (naming reflects the electrode spacing). Four candidate target regions have been defined (shown as colored ellipses). **b** By changing the current ratio, the distribution of the TI modulation amplitude can be steered laterally. **c** shows for both configurations, how changing the ratio affects the averaged TI field of the left (continuous) and right (dashed) target regions for a given total current (graphs are normalized to 2 mA). **d** To translate TI modulation amplitude field to evoked left and right diaphragm responses (evoked ratio) sigmoid/Gaussian curves are fitted to the measured current strength-dependent responses (1:1 current ratio). The data points are the experimental data that the model (thick solid line) is derived from. **e** The deviation (relative standard deviation) between measurements and simulation predictions of the evoked left and right diaphragm responses have been quantified for predictions based on the simulated TI exposure of the four candidate target regions. Motor Pool-based predictions provide the best results. Panel **f** shows, as a function of the current ratio, polar plots comparing the experimentally measured (solid line) left (blue) and right (red) diaphragm evoked ratio to the simulation predictions (dashed line) for a range of total currents and for the two exposure configurations—here only the superior predictions based on 'Motor Pool' exposure are shown.

the carrier frequency); (2) histological verification of neuronal activation (c-fos) unique to TI stimulation, and (3) functional mapping to demonstrate selective movement with stimulation of specific brain regions.

The results of Grossman et al. raise the possibility of treating conditions requiring deep brain stimulation (e.g., Parkinson's disease, depression) using electrodes at the surface of the brain to deliver TI stimulation[6]. The success of such approaches, however, will depend on the ability to use TI stimulation for focal activation of neuronal targets. In this study, we expand the potential reach of TI technology by showing that TI stimulation can effectively activate breathing by targeting the spinal cord. Respiratory control neurons in the medulla and pons are embedded within a complex brainstem network involved with many aspects of autonomic and somatic motor control[30–32]. For this reason, we focused our efforts on targeting spinal respiratory motor neurons, which comes with less potential for serious off target effects (as compared to brainstem targeting) if the interferential currents are not localized to a relatively focal region.

The appeal of the intramuscular TI stimulation approach described here is the rapidity and ease of electrode placement. Adaptation and refinement of this technology for clinical use could provide, for example, the first responder with another tool for rapidly/immediately restoring breathing should other approaches prove ineffective (e.g., bag-mouth valve breathing, naloxone). Naloxone, when available, can be highly effective at restoring breathing after opioid overdose, but is much less effective during dual overdose conditions (e.g., opioids + alcohol, a frequent occurrence). Naloxone also immediately reverses analgesic effects of opioid and can thus leave the first responder with a patient in acute opioid withdrawal stages. By utilizing non-pharmacological means, TI stimulation can restore breathing without the specific need to identify the depressant agent and does not reverse the analgesic effects of opioids while naloxone does, potentially leaving individuals in pain.

Electrical stimulation of the spinal cord, either using epidural or intraspinal electrodes, is rapidly advancing as a means of improving autonomic and somatic motor function following SCI. Intraspinal electrodes can activate specific motor circuits[33–35], and epidural stimulation is often used to provide a non-specific increase in the excitability of neuronal circuits[15,18,36] or targeted activation of sensory afferent pathways[20]. Here, we show that TI stimulation via epidurally placed electrodes provides the ability to steer interferential currents towards regions of interest, similar to the prior description of deep brain stimulation[6]. The ability to steer currents within the spinal cord, as supported by our in vivo and in silico data, represents a potential advantage of TI as compared to square wave epidural stimulation[9]. In our study, cervical spinal cord-directed TI stimulation produced substantial diaphragm activation due to depolarization of phrenic motor

neurons. The phrenic motor neuron pool is in close proximity to the forelimb motor pools, and we often observed concurrent activation of the biceps muscles, albeit at much lower levels compared to the diaphragm. Off-target muscle activation may be mitigated in larger species or with refinements in electrode configuration and/or location to improve the accuracy of current steering, which could be guided by additional modeling experiments.

Local synaptic blockade experiments have advanced the understanding of how TI stimulation is working and confirm TI fields overlap within the immediate vicinity of the phrenic motor pool. We found that TI stimulation-induced diaphragm activation was considerably attenuated by blocking excitatory synaptic inputs in the cervical spinal cord. These results indicate that activation of excitatory presynaptic inputs are a fundamental component of the mechanism by which spinal-directed TI stimulation activates motor neurons. However, excitatory synaptic inputs are not required since the pharmacological blockade experiments show that it is possible for TI stimulation to activate phrenic motor neurons even in the near complete absence of presynaptic input. Thus, the field is likely overlapping within the phrenic motor nucleus and this suggests that TI stimulation may be used as a method to stimulate motor function following SCI in motor pools lacking presynaptic inputs. In silico models of spinal cord stimulation supported this view, indicating that TI stimulation is reaching the ventrolateral spinal cord and the phrenic motor neurons which activate the diaphragm.

In conclusion, we have expanded the scope of applications for TI technology by showing activation of spinal neuronal pathways to restore breathing during hypoventilation or apnea. Further refinement of the technology may provide a new tool for rapidly restoring breathing after drug overdose or other conditions which impair or eliminate the ability spontaneous respiratory muscle activation.

## Methods

**Study design.** The in vivo experiments were performed using adult Sprague Dawley rats ($n = 47$) with procedures that were proved by the University of Florida IACUC. Both male and female rats were studied ($n = 30$ males, $n = 17$ females); the number of each sex is noted in the figure legends. The number of animals for each cohort was determined prior to initiation of the experiments and was based on preliminary experiments not included in the manuscript. The preliminary work indicated a robust impact of TI stimulation on diaphragm activation. This enabled a power calculation suggesting that $n = 4$ animals per group would be sufficient to statistically differentiate the impact of temporal interference for the primary outcome of diaphragm EMG activity. The calculation was done based on an expected difference of means (e.g., stimulation vs. no stimulation) of 75% (arbitrary units), expected standard deviation of 25%, power of 0.9 and alpha of 0.05. Experiments were excluded from analysis if there was an unequivocal mechanical failure (e.g., broken electrode array, stimulator battery failure, heating pad failure). We did not test for outliers in the data, and included all data from each set of experiments. As all in vivo experiments were terminal, the end points were predefined.

Our initial objectives were (1) to determine if intramuscular TI stimulation could restore ventilation in anesthetized rats, and (2) determine if TI stimulation could effectively activate the phrenic motor circuit following spinal cord injury. To answer those questions we needed to determine electrode locations and configurations which would activate the diaphragm with reduced off target (e.g., forelimb muscle activation). To complement these a priori goals, during data collection we decided to perform the set of experiments utilizing intraspinal receptor antagonist to isolate phrenic motor neurons. In addition, we had planned to perform in silico modeling of current-ratio-based TI stimulation steering to evaluate how the maxima of the induced TI modulation amplitude distribution can move in the spinal cord. After reviewing the in silico mathematical modeling data we decided to conduct additional in vivo experiments. Each data point ($n$) represents a single animal, in a controlled laboratory experiment. The primary outcome measures were based on airflow, and diaphragm muscle EMG. We collected additional data including: biceps and intercostal EMG, arterial blood pressure, and arterial blood (partial gas pressures). Further data were collected to monitor the state of the animals such as: peripheral oxygen saturation, end tidal $CO_2$, body temperature. Experimental conditions were alternated between experimental days (e.g., for the intraspinal receptor antagonist experiments: CNQX/AP5 was administered first on day 1 then strychnine/bicuculline was administered first the next day) to reduce variability due to calibration drift or experimental proficiency.

**Statistics and reproducibility**. Detailed explanation of the statistical tests used, are listed in the following sections and within each figure legend. Briefly, the normality and variance of the data was determined to ensure appropriate assumptions were met. If data within a particular experiment did not meet these assumptions non-parametric tests were used. Normality (Shapiro-Wilk) and equal variance (Brown-Forsythe) assumptions and statistical tests were performed in SigmaPlot 14, and confirmed in MATLAB 2019a.

The primary finding of temporal interference-induced activation of the diaphragm muscle to sustain breathing was replicated as follows. We delivered the temporal interference stimulation using two fundamentally different electrode configurations (intramuscular or epidural), and both methods were shown to activate the diaphragm. Further, within the epidural stimulation group, the data were further replicated by testing different electrode configurations. We were able to reproducibility activate the left vs. right diaphragm based on variations in the electrode placement and/or current.

**Generation of TI waveforms**. The kilohertz voltage sinewaves were generated with a Cambridge Electronics Design (CED) Power3 1401 with a DAC conversion rate of 250 kHz, this voltage waveform was logged (100 kS/s) on the 1401 for data analysis. Current was generated using two isolated current source devices (AM Systems, model 2200). When higher currents were required, units were stacked in parallel to produce to isolated waveforms using four stimulus isolators. Sinewaves were ramped and damped at a rate of 0.88 mA/s until the target current was reached. Each source reliably drove current at 5 kHz as verified by a <2% drop in amplitude at 5 kHz vs. 10 Hz, measured with a digital multimeter (3457 A, Hewlett Packard) on 10 kOhm resistor load. When the two current sources were applied to a common conductive load, i.e., a 6 resistor bridge (each 10 kOhm), at 5 kHz and 5.001 kHz the power spectrum density (PSD) at the different frequency was 0.001% of the PSD at the applied kHz frequencies, measured using a FFT spectrum analyzer (SR770, Stanford Research).

**Terminal electrophysiology**. Rats were initially anesthetized with 3% isoflurane and a 24 gauge catheter was placed in the tail vein to enable conversion to urethane anesthesia (1.7 g/kg body weight, infused at 6 ml/h). After urethane infusion a mixture of lactated ringer solution and sodium bicarbonate (4:1) was infused (2 ml/h) to maintain blood pressure and fluid homeostasis. A tracheostomy tube was placed to ventilate (PE 240 tubing) the animal during surgical procedure (50–60% inspired oxygen fraction $O_2$ (FiO$_2$), the inspired $CO_2$ fraction (FiCO$_2$) was adjusted to maintain end tidal $CO_2$ 42–46 mmHg).All animals were left vagal intact to account for the impact of vagal afferents on breathing and cardiovascular regulation. An arterial catheter (PE 50 tubing) was inserted in the right femoral artery to measure blood pressure and allow for sampling of arterial blood.

Electromyographic activity (EMG) was recorded using pairs of 127 µm tungsten wires. Muscles recorded included the left and right mid-costal diaphragm, left 3rd external intercostal, left biceps. EMG activity was amplified (×1000) with a differential amplifier (A–M systems model 1700), filtered 100–1000 Hz, then digitized at 10 or 20 kS/s with a Power 3 1401 (CED, Cambridge UK).

**Experiment 1: Fentanyl overdose with intramuscular TI rescue**. Fentanyl overdose was performed in urethane anesthetized, vagal intact, rats (244–412 g; 9 female/3 male) as described above and ventilated during surgical procedures. To artificially restore ventilation, 127 µm tungsten wires (deinsulated 4 mm from tip and bent into a hook) were inserted intramuscularly. A pair of electrodes was inserted in the muscles on each side of the neck, one dorsal and one ventral (Supplementary Fig. 1). The wire tips were inserted until they approximated the vertebral column near C3.

After EMG and stimulation wire placement, all rats were placed in a prone position and disconnected from the ventilator. A custom 3-D printed pneumotachograph (20 mm long) was placed at the distal end of the tracheostomy tube. The pneumotachograph had a sampling port immediately proximal to the end of the tracheostomy tube which allowed us to sample end tidal $CO_2$ (MicroCapstar, CWE). Two pressure ports separated by a narrow resistor (0.6 mm diameter) were connected to a differential pressure transducer (SDP810, Sensirion). The end of the pneumotachograph opened into a cone with a port, which was connected to tube to provide $O_2$ supplementation. Peripheral capillary oxygen saturation (SPO$_2$) was measured on the left hind foot with a MouseOx Plus (Starr). Oxygen was supplemented to ensure >90% SPO$_2$ (FiO$_2$ 0.5–0.7). We then collected a 10 min baseline period of spontaneous breathing. To model opioid induced respiratory depression we administered fentanyl citrate (30 mcg/kg i.v.) via a tail vein catheter.

Three groups ($n = 4$/group) were studied (1) No stimulation, (2) TI stimulation at 5000 Hz and 5001 Hz to produce a respiratory rate of 60 breaths per minute, (3) TI stimulation with beat frequency matching the endogenous respiratory rate. To accomplish this, we measured the spontaneous respiratory rate (96–116 breaths per minute) during the baseline period and set the offset frequency to match (range: 1.6–1.933 Hz). Stimulation (8–10 mA) began ~1 min after fentanyl administration. The initial stimulation amplitude was determined while the rats were still on the ventilator prior to fentanyl administration and was adjusted to maintain flow rates at or above baseline levels. The stimulation was delivered for 60 s of stable amplitude (plus ramp and damp phases), and was off for 3 s. Stimulation was delivered ($28 \pm 9$ min, mean ± SD; 17 to 40 min, range) until spontaneous respiratory efforts in the off periods produced flow near baseline levels. Animals were euthanized 30 min after stimulation ended. Samples of arterial blood from the femoral catheter were taken at (1) baseline (10 min after switching animals to pneumotachograph), (2) post fentanyl (30–45 s), (3) every 10 min throughout the duration of the prep.

To verify intramuscular electrode placement relative to the spinal segments, we placed 4, 127 µm tungsten wires in the neck of a 233 g male rat as described above. We verified that location of these wires was sufficient to activate the diaphragm with TI stimulation. We then clipped the wires at the surface of the skin and euthanized the rat. Magnetic resonance images were measured with a 4.7 Tesla Oxford magnet, VNMRS console (Agilent Santa Clara, CA), running VnmrJ 3.1, BFG-200/115-S14 gradients (Resonance Research Inc., Billerica, MA), and a 63 mm/108 mm (id/od) Agilent quadrature birdcage H-1 coil. High resolution images, with 35 micron isotropic resolution, were acquired in ~33 min at 200 MHz with a three-dimensional gradient echo pulse sequence using a repetition time of 30 ms, echo time of 10 ms, with 4 signal averages into a data matrix of $128 \times 128 \times 128$. Images were processed with in-house software written in IDL (Harris Geospatial Solutions, Inc., Herndon, VA).

**Experiment 2: Epidural TI stimulation in intact and spinally injured rats**. To determine how best to activate the diaphragm using epidural electrodes 8, vagal intact, rats (222–385 g; all male) were anesthetized as described above and venti-lated during surgical procedures and throughout experiments. Epidural arrays consisting of 30 micron platinum/iridium (80/20%) wires insulated with 5 microns polyimide, glued to a 25 micron polyimide backing were used to deliver the TI field. Arrays were made with either 2 or 6 electrodes arranged in two rows (one lateral, one medial) spaced 1.5 mm apart. 1 mm of each wire was deinsulated by scraping off the polyimide insulation. Initial experiments used a 6 wire grid with a pair of electrodes at C3, C4, and C5 (Fig. 3a–d) in monopolar configurations current was returned to metal retractors in the neck muscles. As C3 monopolar stimulation was determined to be the most optimal later grids only had a pair of electrodes at C3 (Fig. 3f). To determine the effect of return location on diaphragm activation we varied the location of the return (Fig. 3e; ventral, lateral, and dorsal), later experiments used the ventral placement. These initial experiments used a 1:1 current ratio (0.2–1.4 mA/waveform).

To determine if TI could activate the paralyzed diaphragm we studied intact rats ($n = 4$, all male 373–436 g), which then received an acute (<10 min) C2Hx and rats ($n = 4$, all male 366–540 g) with chronic (10 months) C2Hx injuries. To determine the stimulus currents to produce peak diaphragm bursts in the three conditions (Intact, Acute SCI, Chronic SCI), we produced a stimulus response curve by ramping the lateral current (0:60:1800 µA) and stepping up the medial current (0, 60, 180, etc. to 1800 µA). Acute spinal cord injuries were performed immediately after completing the stimulus response curve in intact animals. A micro-knife was used to cut the left side of the spinal cord from midline laterally, sufficient to abolish spontaneous activity ipsilateral to injury, the stimulus response curve was then repeated within 10 min of acute SCI. Surgeries for the chronic spinal cord injuries were performed 10 month prior to study date. The general surgical procedure for the C2Hx surgery has been described[37,38]. Briefly, rats were anesthetized with 2% isoflurane and body temperature was maintained with water recirculating heating pad. The surgical site was shaved and sanitized with alternating chlorhexidine and alcohol scrubs, the skin and muscles were incised in layers and a laminectomy was performed over C2. The spinal cord injury was performed as per the acutely injured animals using a micro-knife. The muscles and skin were sutured together in layers, and the rats recovered in a heated incubator. Buprenorphine (0.03 mg/kg) was given every 12 h for 60 h following surgery, meloxicam (1 mg/kg) was given daily during this time period.

**Experiment 3: Epidural TI current steering in intact rats**. To determine how to steerably activate the diaphragm without moving electrodes, we tested two static electrode configurations in four, vagal intact, rats (275–463 g; 2 female/2 male) which were anesthetized as described above and ventilated during surgical procedures and throughout experiments. An epidural array was fabricated as described above, but with three electrode contacts; one on each side of the cord and one near midline (Fig. 6a). TI was generated on the left side of the spinal cord (lateral and medial, 1.5 mm configuration) or across the cord (left lateral to right lateral; Fig. 6 3 mm configuration). Stimulation was produced in specific ratios 0:1, 1:7, 1:5, 1:4, 1:3, 1:2, 1:1 and their reciprocals such that the sum of currents equaled 1500 to 2400 µA in 300 µA increments, as well as 0 summed current. This allowed us to assess how the focal point (where the fields meet) moved through the tissue as we recorded from the diaphragm, external intercostals, and biceps bilaterally.

**Experiment 4: Epidural TI following intraspinal receptor antagonism**. To assess if TI stimulation acts through presynaptic mechanisms or directly depolarizes phrenic motor neurons to activate the diaphragm, we injected intraspinal receptor antagonists, in 11 anesthetized, vagal intact, rats (231–336 g; 5 female/6 male). Rats were ventilated throughout the entire experiment period. We cut out the center of the epidural stimulation arrays to allow for antagonist injection without having to remove the epidural electrode. Stimulation was delivered at C3. To verify that the receptor antagonist reached the motor pool, we monitored endogenous burst activity prior to each stimulus response curve. The mixture of excitatory receptor antagonists eliminated endogenous bursts on the side of injection (Fig. 5f), without eliminating spontaneous bursts on the contralateral diaphragm. This suggests the injections remained within the spinal cord and is unlikely it reached area of respiratory control in the brainstem. The experimental protocol was as follows: baseline stimulus response curve, antagonist injection, 10 min wait period for diffusion, stimulus response curve, alternative antagonist injection, wait 10 min, final stimulus response curve.

Current on the medial electrode was varied from 100 µA to 2580 µA in steps of 80 µA, while the current on the lateral electrode was held constant at 100 µA, 200 µA, and 300 µA. One group of animals ($n = 5$ 2 f/3 m) were injected with excitatory receptor antagonists first followed inhibitory receptor antagonists, the other group was injected with the alternate order ($n = 5$ 2 m/3 f). Excitatory receptor antagonism was accomplished using a 3 µL injection of a mixture of 2-amino-5-phosphonopentanoic acid (AP5; 12.5 mM, #A8054 and Sigma-Aldrich) and cyanquixaline (CNQX; 500 µM, #C127 Sigma-Aldrich) at C3, C4, and C5, total 9 µL injected. These antagonists blocked NMDA and non-NMDA glutamatergic receptors, respectively[39,40]. Inhibitory receptor antagonism consisted of a 1 µL injection of a mixture of strychnine (400 µM, #S8753 Sigma-Aldrich) and bicuculline (100 µM, #14343 Sigma-Aldrich) at C3, C4, and C5, total 3 µL injected. These antagonists blocked glycine and GABA$_A$ receptors, respectively[41].

**Alternative stimulus waveforms, that do not produce temporal interference**. The primary intent of our work was to evaluate how TI stimulation may be used to restore ventilation during severe hypoventilation, but not to directly compare TI efficacy to other forms of stimulation, however we did examine the impact of some alternative waveforms. The control waveforms consisted of a dual (1) high frequency (5001 Hz), (2) low frequency (1 Hz), or (3) 50 pulses per second (pps) 0.2 ms/phase charge balanced square wave, 0.5 s train duration 1 train/s. The dual high frequency control waveforms (5001 + 5001 Hz) were utilized to verify that the offset (i.e., beat frequency) was required vs. a non-specific excitability of the spinal cord. The dual low frequency waveforms (1 + 1 Hz) were used determine if a high frequency carrier wave was required for diaphragm activation. We used 50 pps epidural square-wave stimulation[42,43] as our final control waveform. We assessed how these waveforms activate the diaphragm, external intercostal and biceps. In urethane anesthetized ventilated, vagal intact, rats with intramuscular electrodes ($n = 3$, 285–306 g, 1 female/2 male) or lateralized (1.5 mm configuration) epidural electrodes ($n = 4$, same rats as in experiment 3), we found the currents sufficient to generate diaphragm bursts with TI stimulation. At the current which produced robust diaphragm activation using TI stimulation, we next generated the control waveforms for 6 s each, while recording EMG activity in the diaphragm, intercostal, and biceps muscles ipsilateral to stimulation.

During a subset of experiments ($n = 3$) using epidural stimulating electrodes, we determined the currents to evoked robust diaphragm activation, then we injected pancuronium bromide (2.5 mg/kg; Hospira) intravenously. This resulted in cessation of endogenous bursting in the diaphragm, following cessation of endogenous bursting reproduced the TI stimulation, which no longer evoked diaphragm bursts. This was performed to validate that the TI stimulation was acting within the nervous system, as pancuronium bromide blocks nicotinic acetylcholine receptors.

**Experiment 5: Comparison of simulated and measured TI evoked respiratory response**. Simulations mimicking epidural experiments were performed using the Sim4Life (ZMT Zurich MedTech AG, Zurich, Switzerland) platform for computational life sciences investigations and were centered on a simplified 2.5D model of the C3 spinal cord tract derived from the Watson Spinal Cord atlas (Fig. 6a). The finite element method (FEM) *Ohmic-Current-Dominated Electro-Quasistatic*

(EQSCD) electromagnetic (EM) solver was used to compute solutions to Eq. (1), where $\sigma$ is the electric conductivity distribution, $\varphi$ the electric potential.

$$\nabla \sigma \nabla \varphi = 0 \qquad (1)$$

The electric field is obtained from Eq. (2).

$$\vec{E} = -\nabla \varphi \qquad (2)$$

This solver is suitable for domain sizes much smaller than the wavelength and when displacement currents are negligible (applicability to the simulated setup was verified). Tissue electric conductivities were assigned according to the low-frequency section of the IT'IS database (17). Electrode wires were placed in accordance with the experiments and treated as perfect electric conductors (PEC). The tissue-entities were discretized using tetrahedral elements (finest mesh resolution: 0.06 mm) with prismatic shell elements inserted between the cerebrospinal fluid (CSF) and the epidural fat to model the semi-insulating dura. Dirichlet (voltage) boundary conditions were assigned to the electrode surfaces providing and input current of 1 mA. Two EM simulations were conducted for each electrode configuration, one for each wire pair. For a given set of steering parameters (current magnitudes), the fields were correspondingly rescaled and maps of maximum amplitude modulation (MAM) calculated according to Eq. (3) that is from Grossman et al.[6].

$$|TI(w_1, w_2)| = \begin{cases} 2\left|\overrightarrow{E_{2w}}(\vec{r})\right| & \text{if } \left|\overrightarrow{E_{2w}}(\vec{r})\right| < \left|\overrightarrow{E_{1w}}(\vec{r})\right|\cos(\alpha) \\ 2\left|\overrightarrow{E_{2w}}(\vec{r}) \times \left(\overrightarrow{E_{1w}}(\vec{r}) - \overrightarrow{E_{2w}}(\vec{r})\right)\right| / \left|\overrightarrow{E_{1w}}(\vec{r}) - \overrightarrow{E_{2w}}(\vec{r})\right| & \text{otherwise} \end{cases}$$

$$(3)$$

To compare simulated and measured evoked ratios of the epidural experiments, the average MAM in four target regions (elliptic prisms covering the 'Motor Pool', 'Intermediate', 'Bulbospinal', and 'Afferent' regions—see Fig. 6a—and extending over the length of the exposed wire) was computed for varying steering parameters (in accordance to the experiments) and converted to predicted evoked ratios according to transfer functions (evoked ratio as a function of current strength). These transfer functions have been obtained through fitting of all experimental data available for configurations with equal channel currents (1:1 ratios, Fig. 3d) and they permit to relate MAM exposure strength to the evoked response. These same methods were used to predict the effect of return location (Fig. 3e). Sigmoidal fits were used for the prediction of left diaphragm evoked ratio (one for the '1.5 mm' and one for the '3 mm' electrode condition). For symmetry reasons, the transfer function of the right diaphragm in the '3 mm' electrode configuration was assumed to be identical to that of the left diaphragm (accordingly, the measurement data of the left and right diaphragm evoked responses were pooled). The right diaphragm response for that condition shows an unexpected maximum at 2 mA. Therefore, a shifted Gaussian was used for fitting, instead of a sigmoid. The transfer functions were kept constant, independent of the steering parameters. The differences (standard deviations) between experimental data and simulation predictions can be seen in Fig. 6e.

Assumptions and simplifications: The computational model of epidural stimulation is clearly highly simplified, with important uncertainties and simplifications related to anatomical detail and accuracy, the tissue property assignment, the quality of electrode contacts, *etc*. The following are some of the most important ones to keep in mind when comparing modeling results to experimental data:

TI metric: The formula from Grossman et al. has been used to quantify exposure[6]. It is related to the maximum amplitude of the modulation envelope along any orientation. However, (i) while modulation envelope amplitude explains the observed steering behavior, it has not strictly shown to be the relevant exposure quantity, (ii) if the relevant neurons show strong preferential orientation, the magnitude of the projection of the modulation envelope could dominate, and (iii) more than one exposed population are likely involved in the response and they are part of a network, which can prevent simple mapping of local TI exposure strength to physiological response.

Electrode placement: There is a large degree of uncertainty associated with the exact electrode placement, which cannot even be resolved by the available imaging data.

Current ratio: The steering is determined by the ratio of the currents applied to the two channels[6]. Any mismatch between the experimental setup and the computational model that affects the impedance (e.g., electrode contact) will therefore result in an error of the predicted steering.

CSF: The high electric conductivity of CSF means that intra-spinal fields are strongly affected by the distribution of CSF and its ability to route current around the spinal cord. The amount of pressure exerted by the electrode and the resulting CSF displacement can therefore have a major impact.

Spinal dorsal roots: The thin and densely packed spinal roots are not considered in the computational model, but they are frequently a primary target of spinal-cord stimulation[44]

The variability associated with the measurement data illustrates that there is an important degree of uncertainty associated with biological response, intra-subject anatomical variability, and/or electrode placement reproducibility.

**Data analysis**. Normality (Shapiro-Wilk) and equal variance (Brown-Forsythe) assumptions and statistical tests were performed in SigmaPlot 14. If either test

failed, the appropriate nonparametric test was used, all tests performed were two-sided. In vivo electrophysiology data was analyzed using MATLAB 2019a. Respiratory airflow signals were rectified and integrated (4 s time constant) to form average airflow traces. To assess the impact of TI rescue in fentanyl overdose on airflow and mean arterial pressure we used separate two-way ANOVAs (Group x Time; Fig. 2).

Evoked muscle activity was band passed filtered (100–1000 Hz, 2-pole Butterworth), then infinite impulse response (iir)-notch filtered (5000 Hz center, 300 Hz bandwidth). Filtered EMG signals were then rectified and a moving median filter (50 ms time constant) was performed to reduced ECG contamination, finally the signal was integrated (50 ms time constant). To determine if the EMG activity was evoked during the peak of the TI envelope, we calculated the timing of the TI envelope by summing the high frequency voltage waveforms (stimulation sum), rectifying and integrating (50 ms time constant). We then calculated an evoked ratio of EMG activity during the peak of the TI envelope over the trough of the TI envelope (Fig. 3b). A one-way ANOVA on ranks was performed to assess which electrode configuration produced the largest normalized evoked ratio (Fig. 3d). We also calculated peak EMG during TI envelope to assess how spinal cord injury affects the ability to activate the diaphragm. We determined the current ratio which evoked the largest peak diaphragm EMG in the three conditions (Intact–Lat 1320:600 µA Med, Acute SCI–Lat 360:1080 µA Med, Chronic SCI–Lat 360:600 µA Med), then assessed the peak EMG output in the diaphragm at those ratios for all three conditions. A one way ANOVA found no differences in evoked burst amplitude (Fig. 3i). To assess the impact of intraspinal receptor antagonism we performed cycle triggered averaging of the ipsilateral diaphragm bursts, using the contralateral diaphragm bursts (Fig. 5D–F). To assess the impact of receptor antagonism we compared evoked burst amplitudes for three lateral currents (100–300 µA) with medial current standardized at the value which evoked diaphragm EMG activity 25% above endogenous (pre-drug) baseline bursting using a one-way ANOVA or one-way ANOVA on ranks if the data were non-parametric (Fig. 5h, i).

**Reporting summary**. Further information on research design is available in the Nature Research Reporting Summary linked to this article.

## Data availability

The data that support the findings of this study are available from the corresponding author upon reasonable request.

## Code availability

Custom codes written to analyze the data within this manuscript are available from the corresponding author upon reasonable request.

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

## Acknowledgements

This work was supported by funding from the National Institutes of Health, grant numbers: F31HL145831 (M.D.S.), R21NS109571 (D.D.F.). A portion of this work was performed in the McKnight Brain Institute at the National High Magnetic Field Laboratory's Advanced Magnetic Resonance Imaging and Spectroscopy Facility, which is supported by National Science Foundation Cooperative Agreement No. DMR-1644779 and the State of Florida. We thank Dr. Elisa Gonzalez-Rothi for her assistance with surgical procedures and Dr. Huadong Zeng for his assistance in MR imaging. We also thank Drs. Niels Kuster and Sabine Regel for thoughtful comments on the manuscript.

## Author contributions

All authors interpreted data, reviewed, and approved this work. M.D.S. collected and analyzed all in vivo data, drafted the manuscript and in collaboration with co-authors conceived, designed, and interpreted the work. A.M.C. and E.N. developed computational model and suggested complementary in vivo studies N.G., K.J.O., and E.S.B. helped design experiments and interpret data. T.H.M. aided in collecting MRI data and provided insight on experiments. D.D.F. designed and conceived of experiments, interpreted data, and provided substantial work on the text.

## Competing interests

N.G. and E.S.B. have a patent on TI technology, assigned to MIT. E.N. is a board member and shareholder of TI Solutions AG, which develops stimulation devices and treatment-planning tools for temporal interference research. M.D.S., K.J.O., and D.D.F. have submitted a patent application related to use of TI for stimulating breathing. T.H.M. and A.M.C. declare no competing interests.
