## [Peer Review File · Communications Biology]

Reviewers' comments:

Reviewer #1 (Remarks to the Author):

This is an important and exciting study demonstrating the potential for TI stimulation paradigms to rescue ventilatory depression following opioid overdose or high cervical spinal injury. There is a pressing need to develop alternative strategies to sustain breathing during opioid overdose since current strategies are inadequate and may induce opioid withdrawal symptoms. Similarly, there is a need to develop strategies to enable independent breathing following spinal injury. The investigators convincingly demonstrate that TI stimulation can be used to induce diaphragm activation and restore ventilation to levels that can at least partially restore normal blood gases. This study is very well controlled and highly novel. Overall, I am highly enthusiastic about this study and feel it has the potential to transform the way clinicians treat individuals with respiratory depression.

Abstract:

1) Second to last line, correct "moto neurons"

Introduction:

2) Line 43: "Psychiatric disorders" seems to come out of nowhere. I recommend expanding on this thought or deleting altogether

Figures/Results:

3) Figure 1A is very difficult to see because it is too small.

4) If I understand correctly, the investigators first gave TI stimulation to a spontaneously breathing rat. It appears that the natural breathing rhythm became entrained to the stimulation parameters (Figure 1C). Was the vagi intact in these animals? Please comment.

5) Figure 1: Please quantify left/right hemidiaphragm activation during and after TI stimulation. Although it is useful to know that the diaphragm was activated in 4/4 rats, this doesn't reveal how well the diaphragm was recruited.

6) Lines 100-102: "To verify this effect we modeled the impact of the location of the return electrode in accordance with the experimental data shown in Fig. 3E. This modeling produced a variability of $\pm 3\%$, and thus the simulation results also indicate that the location of the return electrode had minimal impact on diaphragm activation." I was unable to find a description of how these data were modeled, but perhaps I overlooked it in the manuscript.

7) Figure 3H: please show blood pressure responses

8) Was the steering study performed in spontaneously breathing, spinally intact rats? I am a little confused as to why there is no spontaneous diaphragm activity in the absence of stimulation.

9) Did the steering protocol also result in activation of the biceps and intercostals, in addition to diaphragm? Would this limit the utility of TI stimulation in SCI?

10) Line 125: "Fig. SA" should be changed to "Fig. S3"

11) Line 131: "Rhythmic diaphragm activation increased when the current ratio shifted to 1:2, but then ceased as the current ratio continued to shift." While the average data support this statement, the representative trace in Figure 4A does not since diaphragm activity continued to increase or plateaued up to $\sim 1:5$ (this is stated in the legend, and appears to contradict the manuscript text). Either make clear that the manuscript text is referring to the average data and/or soften the statement to "in most animals". Similarly, the legend states that "When the medial current was higher than the lateral (greater than 1:1 ratio) neither side of the diaphragm was activated." Again, the representative trace seems to contradict this statement, as do 2/4 rat responses.

12) Average data Figure 4A and B: far left label, do the authors mean "1:0"?

13) Lines 158-162: a range of % changes are given for the evoked diaphragm stimulation responses after pharmacological inhibition of excitatory or inhibitory neurotransmission, but it isn't clear where these values come from. Are these the average data for the 3 different lateral currents, or individual responses? Neither seems to correspond with the data presented in H and I.

14) Please show the data from panB and 50 pps experiments.

15) Figure S2: legend does not match bar colors in graph. In the post-stimulation period, presumably after fentanyl has worn off and normal spontaneous breathing has resumed, rats that were driven to breathe at 60 bpm are still fairly hypercapnic. Please comment.

Discussion:

16) A section on how the authors think TI stimulation results in ipsi-hemidiaphragm activation following SCI is warranted – i.e., is it stimulation of crossed pathways? Or directly depolarizing phrenic motor neurons? In uninjured rats, the authors found that TI-induced diaphragm activity was largely due to activation of pre-synaptic inputs -- would these mechanisms be different in the injured spinal cord?

17) TI stimulation appears to activate other, non-respiratory motor neuron pools. Does this limit the potential utility of TI stimulation to only emergency situations?

Methods:

17) Line 378: Please clarify "one lateral on medial"

18) How soon after acute SCI were animals studied?

19) Line 406: Only current ratios of 1:1 to 1:9 were mentioned, but higher ratios were also studied (ie., 1:0, 7:1, 5:1, etc).

Reviewer #2 (Remarks to the Author):

The investigators present a study using HF temporal interference electrical stimulation to restore diaphragm activity in opioid-induced apnea and dorsally placed electrodes in a rat spinal cord injury model. The concept is novel and the application to respiratory compromise is very interesting and potentially important. The investigators are experienced and well placed to carry out this type of research. The interest of this study lies in the concept of how TI stimulation can be used to stimulate deep structures and there are a few questions regarding potential clinical applications. The results are impressive.

Comments;

1. For acute opioid treatment, what is the advantage of using electrical stimulation when a safe and robust method would be to use a bag and mask? I accept that naloxone may have variability in how long it takes to work, but would not inserting wires into the neck also have a greater risk and require technical skill?

2. In figure 3D, stimulation at B1, B2 and B3 also activated biceps and external intercostal muscles. In the potential human SCI patient, would this unwanted stimulation not be a problem? It would seem that there is a greater differential at A2 and A3 between diaphragm and other muscles but does the stimulation at these areas produce useful diaphragmatic stimulation?

3. Fig 2C – I am intrigued by the restoration of MABP with phrenic nerve stimulation. If the stimulation is working simply by 'pacing' the diaphragm, what is the mechanism for rise in MABP? One possibility is related to the Hering-Breuer reflex but it seems like quite an exaggerated response and I wonder if it could be something else?

Alex Green

Response to reviewer comments.

We thank the reviewers for their comments. We have addressed each comment in the revised manuscript, and a detailed response is provided next:

Reviewer 1

Comment: Several typos were identified

Response: Corrected

Comment: “Psychiatric disorders” seems to come out of nowhere. I recommend expanding on this thought or deleting altogether

Response: The mention of TI for psychiatric disorders has been removed.

Comment: Figure 1A is very difficult to see because it is too small.

Response: We increased the overall size of Figure1 and increased the relative proportion of panel A.

Comment: The reviewer asked to clarify if the vagus nerve were intact during the experiments.

Response: All animals in this study were left vagal intact, this has been clarified throughout the text. Due to the powerful impact of vagal sensory afferents on breathing and cardiovascular regulation we felt it was best to leave the vagus nerves intact during the TI experiments.

Comment: Please quantify the extent of hemidiaphragm activation during and after TI stimulation.

Response: Thanks for this suggestion. For the revised submission we quantified the magnitude of diaphragm activation and these data are now included in the results section (see page 5, lines 53-55).

Comment: Missing/unclear description of modeling performed for Figure 3E.

Response: We modeled the same locations for the return electrode that were tested *in vivo*, and no significant impact on the TI modulation envelope distribution was apparent. For quantification, the same approach as in Fig. 6C was used. The manuscript has been modified in the methods sections (page 34, lines 498-500).

Comment: Figure 3H: please show blood pressure responses

Response: Thanks for pointing out this omission. We now include arterial blood pressure on the example of the response to epidural TI stimulation (panel H bottom trace, Figure 3, page 12).

Comment: Was the steering study performed in spontaneously breathing, spinally intact rats? I am a little confused as to why there is no spontaneous diaphragm activity in the absence of stimulation.

Response: Yes, the TI steering experiments were performed in mechanically ventilated, vagal intact, and spinal-cord intact rats. The amplitude of the endogenous bursts was relatively low during the baseline condition, and this served to mitigate the impact of the endogenous burst on the evoked ratio calculation. The small degree of spontaneous diaphragm EMG activity is now highlighted in the figure, with an inset, and associated legend.

Comment: Did the steering protocol also result in activation of the biceps and intercostals, in addition to diaphragm? Would this limit the utility of TI stimulation in SCI?

Response: Yes, the TI stimulation does activate other motor pools as show in Figure 3D and Supplemental Figure S5. In this regard, there are several important points:

- Our primary goal was to determine the viability of spinal-directed TI stimulation, and not to achieve 100% specificity of target motor pool activation. The mathematical modeling data (Figure 6) provide evidence that the TI electrical currents are hitting the ventral horn of the spinal cord, and we plan to continue developing the TI technology to further reduce off target effects on limb muscles. We were very encouraged that the motor impact of cervical epidural TI was primarily the diaphragm, and autonomic

functions were not impacted. As a final consideration, off target activation is likely to be less in species with a larger spinal cord.

- The relative activation of “off-target” muscles was considerably less than the observed diaphragm activation during TI (see Figs. 3D and S5).
- When compared to off target activation during “traditional” square wave spinal cord stimulation, the relative extent of off target muscle activation was considerably lower with TI stimulation (Fig. S5).
-

Comment: Line 131: “Rhythmic diaphragm activation increased when the current ratio shifted to 1:2, but then ceased as the current ratio continued to shift.” While the average data support this statement, the representative trace in Figure 4A does not since diaphragm activity continued to increase or plateaued up to ~1:5 (this is stated in the legend, and appears to contradict the manuscript text). Either make clear that the manuscript text is referring to the average data and/or soften the statement to “in most animals”. Similarly, the legend states that “When the medial current was higher than the lateral (greater than 1:1 ratio) neither side of the diaphragm was activated.” Again, the representative trace seems to contradict this statement, as do 2/4 rat responses.

Response: We agree. Thank you for helping us clarify the data presentation. Please see pages 14-15, lines 129-145 for a revised summary of the data.

Comment: Average data Figure 4A and B: far left label, do the authors mean “1:0”?

Response: Yes, we have corrected this in the figure.

Comment: Lines 158-162: a range of % changes are given for the evoked diaphragm stimulation responses after pharmacological inhibition of excitatory or inhibitory neurotransmission, but it isn’t clear where these values come from.

Response: These ranges were in reference to the effect of receptor antagonism across all stimulation currents. The text has been modified to mention the specific effects shown in the figures.

Comment: Please show the data from panB and 50 pps experiments.

Response: The impact of pancuronium bromide on the TI response is now provided in Figure S4. In addition, a detailed example of the response to square wave (50 and 300 pps) vs. TI stimulation is shown in Figure S5.

Comment: Figure S2: legend does not match bar colors in graph.

Response: Fixed.

Comment: In the post-stimulation period, presumably after fentanyl has worn off and normal spontaneous breathing has resumed, rats that were driven to breathe at 60 bpm are still fairly hypercapnic.

Response: Yes, and in fact this is to be expected based on our laboratories experience. This hypoventilation is due to the urethane anesthesia and the fact that the rats had not been mechanically ventilated for > 1 hour at that point of the experiment. The primary point we are making is that the TI stimulation enabled the survival of the animals. If the experiments had been continued to the point that urethane had “worn off” (which was not permitted, per IACUC), it can be reasonably expected that the hypercapnia would have resolved.

Comment: A section on how the authors think TI stimulation results in ipsi-hemidiaphragm activation following SCI is warranted – i.e., is it stimulation of crossed pathways? Or directly depolarizing phrenic motor neurons? In uninjured rats, the authors found that TI-induced diaphragm activity was largely due to activation of pre-synaptic inputs -- would these mechanisms be different in the injured spinal cord?

Response: Thank you for giving us license to speculate on this issue. We have elaborated on these topics in the discussion (pages 25-16, lines 285-296).

Comment: TI stimulation appears to activate other, non-respiratory motor neuron pools. Does this limit the potential utility of TI stimulation to only emergency situations?

Response: This is certainly a possibility, and we acknowledge this in the revised text (page 25, lines 281-285). However, the mathematical modeling data (Figure 6) provide evidence that the TI electrical currents

are hitting the ventral horn of the spinal cord, and we plan to continue developing the TI technology to further narrow the stimulus energy to the phrenic motor pool. We were very encouraged that the motor impact of cervical epidural TI was primarily the diaphragm, and autonomic functions were not impacted. As a final consideration, off target activation is likely to be less in species with a larger spinal cord.

Comment: How soon after acute SCI were animals studied?

Response: The rats were studied within 10 minutes of acute SCI, and the text has been revised to clarify this.

Comment: Only current ratios of 1:1 to 1:9 were mentioned in the methods, but higher ratios were also studied (ie., 1:0, 7:1, 5:1, etc).

Response: Thanks for pointing this out. All the current ratios are now listed in the methods.

Reviewer 2

Comment: For acute opioid treatment, what is the advantage of using electrical stimulation when a safe and robust method would be to use a bag and mask? I accept that naloxone may have variability in how long it takes to work, but would not inserting wires into the neck also have a greater risk and require technical skill?

Response: This is certainly a fair point. For this study, we are not attempting to say that bag masks are obsolete or ineffective, but rather we are introducing an approach that with further refinement may be an effective “rescue” treatment for hypoventilation. We feel that this is a unique demonstration of a new technology that very effectively stimulates breathing by activating the diaphragm muscle. Our hope that further refinement of TI stimulation may allow for the stimulation to be delivered in a manner that is even less invasive (e.g., surface electrodes) and more selective (e.g., stimulus energy focused exclusively on phrenic motor neurons). One could envision certain situations where the bag/mask is not effective (e.g., single responder, multiple overdose victims), or having a TI-based system in place as a back-up should the bag/mask not be effective. Thus, we present a new technology and hope that others in the field benefit and continue working with TI to improve its use.

Comment: In Figure 3D, stimulation at B1, B2 and B3 also activated biceps and external intercostal muscles. In the potential human SCI patient, would this unwanted stimulation not be a problem? It would seem that there is a greater differential at A2 and A3 between diaphragm and other muscles but does the stimulation at these areas produce useful diaphragmatic stimulation?

Response: Yes, off-target activation can certainly be a problem. Here we have demonstrated the viability of spinal-directed TI stimulation for activating the diaphragm after SCI, but have not shown 100% specificity for activating only the mid-cervical phrenic motor neuron pool. In this regard, the mathematical modeling data (Figure 6) are informative and provide evidence that the TI electrical currents are hitting the ventral horn of the spinal cord. We were very encouraged that the motor impact of cervical epidural TI was primarily (albeit not exclusively) on the diaphragm, and autonomic functions were minimally impacted. Our research team is now focused on the continued development of the TI technology to further reduce off target effects on limb muscles. By publishing this initial work, we hope to alert others in the field to promise of TI so that other laboratories can continue to help refine the method. As a final consideration, off target activation is likely to be less in species with a larger spinal cord.

Regarding the A2 vs. A3 differential: While the “A” configurations differential effect between the diaphragm and biceps, they did not allow for robust diaphragm activation. Further, the “A” configurations are not as conducive to steering as the electrodes are not distributed around the circumference of the spinal cord. We have added additional text in the discussion addressing the potential limitations.

Comment: If the stimulation is working simply by ‘pacing’ the diaphragm, what is the mechanism for rise in MABP? One possibility is related to the Hering-Breuer reflex but it seems like quite an exaggerated response and I wonder if it could be something else?

Response: Based on the temporal dynamics of the response, our hypotheses is that the rise in MABP after TI occurs as a (positive) consequence of the restoration in airflow. In other words, the restoration of breathing (and oxygenation) is likely to have produced the rise in MABP. The increase in MABP occurred gradually after starting TI stimulation, and never occurred without concurrent activation of the diaphragm. We

think that this indicates that the increase in blood pressure is not due to activation of pre-sympathetic ganglia, but rather is due to increased breathing.